



# Combined UV and IR ozone profile retrieval from TROPOMI and CrIS measurements

Nora Mettig[1], Mark Weber[1], Alexei Rozanov[1], John P. Burrows[1], Pepijn Veefkind[2], Nadia Smith[3], Anne M. Thompson[4,5], Ryan M. Stauffer[4], Thierry Leblanc[6], Rigel Kivi[7], Matthew B. Tully[8], Roeland Van Malderen[9], Ankie Piters[2], Bogumil Kois[10], René Stübi[11], and Pavla Skrivankova[12]

[1]Institute of Environmental Physics, University of Bremen, Bremen, Germany
[2]Royal Netherlands Meteorological Institute (KNMI), De Bilt, Netherlands
[3]Science and Technology Corporation, Columbia, MD, USA
[4]NASA/Goddard Space Flight Center, Greenbelt, MD, USA
[5]University of Maryland-Baltimore County/Joint Center for Earth Systems Technology, Baltimore, MD, USA
[6]Jet Propulsion Laboratory, California Institute of Technology, Wrightwood, CA, USA
[7]Finnish Meteorological Institute, Space and Earth Observation Centre, Sodankylä, Finland
[8]Bureau of Meteorology, Melbourne, Australia
[9]Royal Meteorological Institute of Belgium, Brussels (Uccle), Belgium
[10]Institute of Meteorology and Water Management, Warsaw, Poland
[11]Federal Office of Meteorology and Climatology MeteoSwiss, Payerne, Switzerland
[12]Czech Hydrometeorological Institute, Praha, Czech Republic

**Correspondence:** Nora Mettig (mettig@iup.physik.uni-bremen.de)

**Abstract.**

Vertical ozone profiles from combined spectral measurements in the ultraviolet and infrared spectral range were retrieved by using data from TROPOMI/S5P and CrIS/Suomi-NPP, which are flying in loose formation three minutes apart in the same orbit. A previous study of ozone profiles retrieved exclusively from TROPOMI UV spectra showed that the vertical resolution

in the troposphere is clearly limited (Mettig et al., 2021). The vertical resolution and the vertical extent of the ozone profiles is improved by combining both wavelength ranges compared to retrievals limited to UV or IR spectral data only. The combined retrieval particularly improves the accuracy of the retrieved tropospheric ozone and to a lesser degree stratospheric ozone up to 30 km. An increase in the degree-of-freedom by one was found in the UV+IR retrieval compared to the UV-only retrieval. Compared to previous publications, which investigated combinations of UV and IR observations from the pairs OMI/TES and

GOME-2/IASI, the degree of freedom is lower, which is attributed to the reduced spectral resolution of CrIS compared to TES or IASI. Tropospheric lidar and ozonesondes were used to validate the ozone profiles and tropospheric ozone column (TOC). From the comparison with tropospheric lidars both ozone profiles and TOCs show smaller biases for the retrieved data from the combined UV+IR observation than the UV observations alone. While the TOCs show good agreement, the profiles have a positive bias of more than 20% between 10 and 15 km. The reason is probably a positive stratospheric bias from the IR retrieval.

The comparison of the UV+IR and UV ozone profiles up to 30 km with MLS (Microwave Limb Sounder) demonstrates the improvement of the UV+IR profile in the stratosphere.



# 1 Introduction

The accurate observation of the vertical distribution of ozone is essential to assess the recovery of the stratospheric ozone layer following the measures taken to phase out ozone depleting substances (ODS) by the Montreal Protocol in 1987 and its Amend-
ments (World Meteorological Organization, 2018). In order to assess the role of tropospheric ozone for air quality (Lefohn et al., 2018) and climate change (IPCC, 2021) accurate measurements of tropospheric ozone are required as well. While sparse in-situ measurements, such as ozonesondes and ground-based lidar ozone profiles have a higher accuracy and vertical resolution, passive remote sensing instrument observing in nadir provide near global coverage. However, the vertical resolution of the ozone profiles from nadir satellite measurements is poorer. Ozone profile retrievals using different satellite measurement tech-
niques (solar/lunar/stellar occultation, limb and nadir) in different spectral wavelength ranges, e.g. ultraviolet (UV), infrared (IR) and microwave, have been developed and evolved over the past decades. Rayleigh scattering and the ozone absorption in its Hartley (200 – 310 nm) and Huggins (310 – 350 nm) bands result in the penetration of UV radiation being strongly wavelength dependent. As first pointed out by Singer and Wentworth (1957), this provides an opportunity to determine vertical profiles of ozone, when observing the UV up-welling radiance from satellite platforms. This approach has been exploited by
the following instruments: NASA Backscatter Ultraviolet (BUV) on Nimbus 4 (Robinson, 1970) and SBUV (Solar BUV) on Nimbus 7 (Heath et al., 1975) on the series of NOAA polar platforms (Cebula et al., 1998), ESA GOME on ERS-2 (Burrows et al., 1999), SCIAMACHY on Envisat (Noel et al., 1998), GOME-2 (Munro et al., 2016) on ESA/EUMETSAT Metop-A to -C, OMI on NASA Aura (Levelt et al., 2006) and TROPOspheric Monitoring Instrument (TROPOMI) on Sentinel-5 Precursor (S5P) (Veefkind et al., 2012).

Starting with the Global Ozone Monitoring Experiment (GOME) instrument, vertical ozone profiles from the troposphere up to the higher stratosphere were retrieved using highly resolved and continuous spectra in the UV (Hoogen et al., 1999; Hasekamp and Landgraf, 2001). One focus has been on improving tropospheric ozone, with Liu et al. (2005) highlighting the importance of extensive spectral corrections needed before a retrieval is possible. Ozone profiles were also retrieved from the successor instruments GOME-2 aboard the series of MetOp platforms (Miles et al., 2015) and used to generate contiguous time
series from multiple instruments (van Peet et al., 2014). Long term analysis of ozone profiles is also possible with the Ozone Monitoring Instrument (OMI) instrument (Huang et al., 2017), which was launched on Aura in 2004 and is still working today. After an extensive re-calibration, it is possible to determine profiles in the stratosphere and tropospheric ozone with an accuracy of up to 10% (Liu et al., 2010b, a). Through validation with ozone sensors and lidar measurements, similar results were also obtained for the UV measurements from the TROPOspheric Monitoring Instrument (TROPOMI) on Sentinel-5 Precursor (S5P)
(Mettig et al., 2021).

While the major challenge for the profiles from UV measurements is the low vertical resolution in the altitude range below 20 km, ozone profiles from IR measurements provide more information about the troposphere, but typically do not retrieve ozone above about 30 km (Bowman et al., 2002). IR ozone profile retrievals use the atmospheric emission in the thermal infrared (TIR) spectral range within the 9.6 μm ozone absorption band. Vertical ozone profiles and tropospheric ozone were
derived from Tropospheric Emission Spectrometer (TES) on Aura (Bowman et al., 2006; Worden, 2004) and from Infrared



Atmospheric Sounding Interferometer (IASI) on Metop-A,-B and -C (Eremenko et al., 2008; Boynard et al., 2009). Together with the Advanced Technology Microwave Sounder (ATMS), the Cross-track Infrared Sounder (CrIS) on Suomi National Polar-orbiting Partnership (SNPP) provides temperature and many trace gas profiles. For the ozone profile retrieval, an overall accuracy of 10% and a precision of 20% up to 35 km are possible using CrIS IR measurements (Nalli et al., 2018). However,
CrIS's vertical resolution is limited and only 1.9 degree of freedom (DOF) corresponding to the information content of two atmospheric layers can be achieved (Smith and Barnet, 2019, 2020).

Combining UV and IR spectral measurements from different instruments improves the information content of ozone profile retrievals providing a high vertical resolution in the stratosphere up to 55 km determined by the UV region and a high vertical resolution in the troposphere from using the IR range. The concept of using UV and TIR observations to improve the
vertical profile of ozone was first discussed in the geostationary tropospheric pollution explorer (GeoTROPE) mission concept (Burrows et al., 2004). The improvement in tropospheric ozone was shown for several combination of instruments: simulated OMI and TES measurements (Landgraf and Hasekamp, 2007), real OMI and TES measurements (Worden et al., 2007a; Fu et al., 2013), GOME-2 and IASI (Cuesta et al., 2013; Costantino et al., 2017; Cuesta et al., 2018) and OMI together with AIRS (Fu et al., 2018). From validation with ozonesondes in the studies with OMI+TES and GOME-2+IASI it was found, that the
relative mean bias and the RMS of the combined ozone profile retrieval is reduced in comparison to the UV only retrieval. However, the IR only retrieval is better than the combined approach in the troposphere. For GOME-2+IASI, an increase of total DOF from 3.3 DOF (for both, UV and IR) to 5 DOF was found, of which 1.6 DOFs are in the troposphere (<12 km) (Cuesta et al., 2013). Using OMI and TES, 6.8 DOF were achieved for the entire atmosphere (UV: 5.5 DOF, IR: 4.3 DOF) with around 2 DOF below 20 km (UV: 1 DOF, IR: 1.7 DOF).

Here we present ozone profiles retrieved from combined TROPOMI UV and CrIS IR measurements. For both instruments individually ozone profiles have been successfully retrieved (Mettig et al., 2021; Barnet, 2019a) but their measurements have not been combined so far. We show and discuss the capabilities and limits of the combined retrieval, present some diagnostics and validate the results by comparisons with ozonesondes and lidars. The main difference from earlier combined UV-IR retrievals is the lower spectral resolution of the IR part (CrIS). The infrared spectrometers TES and IASI have a better spectral
resolution: TES $0.1$ cm$^{-1}$ and IASI $0.25$ cm$^{-1}$ compared to $0.625$ cm$^{-1}$ for CrIS. The question to be answered is whether and to what extent an improvement of the vertical ozone profile retrieval can be achieved in combination with CrIS.

## 2 Data

### 2.1 TROPOMI

TROPOMI (TROPOspheric Monitoring Instrument) is a nadir-viewing ultraviolet and visual spectrometer aboard the S5P
satellite (Sentinel-5 Precursor). It was launched in October 2017 as part of the Copernicus Programme and was supposed to close the gap between the past Envisat (until 2012), the current OMI/Aura, and the future Sentinel-5 spacecraft (launch next year). S5P moves in a sun-synchronous orbit with an equatorial crossing time of 13:30 local time. The instrument provides measurements in the UV ($270 - 330$ nm), UVIS ($320 - 500$ nm), NIR ($675 - 775$ nm) and SWIR ($2305 - 2385$ nm) spectral





channels (Veefkind et al., 2012). For the ozone profile retrieval only the UV1 (270 – 300 nm) and UV2 (300 – 330 nm) radiance

channels are used. Both channels have a spectral resolution of 0.5 nm and a sampling of 0.065 nm. The spatial resolution depends on the channel and on the position in the swath. At the nadir-viewing points it is $28.8 \times 5.6$ km$^2$ (cross-$\times$along-track) in UV 1 and $3.6 \times 5.6$ km$^2$ in UV2. The smaller TROPOMI pixels are binned together to match the coarser spatial resolution of CrIS.

   TROPOMI, like other instruments of this type, shows drift and degradation effects in the UV channels and needs an extensive

pre- and post-launch calibration (Ludewig et al., 2020).For this study, we use the Level 1B version 2 data. In our UV-only retrieval additional calibration steps as part of the profile retrieval is needed Mettig et al. (2021). The version 2 data set is limited to 12 weeks distributed over the period from July 2018 to October 2019. Especially in the lower UV range, the measured intensities have rather low signal-to-noise ratios. In addition to the quality parameters provided by the data sets, we only use UV1 pixels with a mean SNR greater than 20 and UV2 pixel with a mean SNR greater than 50.

## 95 2.2 IR: CrIS

CrIS (Cross-Track Infrared Sounder) aboard Suomi-NPP is a Fourier-Transform spectrometer which provides soundings in the thermal IR spectral range. Suomi-NPP moves in the same orbit as S5P in a loose formation with TROPOMI. The time difference between the measurements from both instruments above the same location is around three minutes. CrIS covers three IR wavelength ranges with 2,211 spectral points: LWIR (9.14 – 15.38 μm), MWIR (5.71 – 8.26 μm), and SWIR (3.92 –

4.64 μm) (Han et al., 2013; Strow et al., 2013; Tobin et al., 2013). The spectral resolution is 0.625 cm$^{-1}$, which is coarser than from instruments like TES or IASI. But in comparison to IASI, CrIS has a lower noise. Hence, the ozone information content depending on both, spectral resolution and noise, it should be similar for CrIS and IASI. For this study we use a spectral window in LWIR between 9.35 and 9.9 μm from the level 2 CLIMCAPS full spectral resolution cloud cleared radiances V2 data product (Barnet, 2019b). The ozone profile used in the validation and the surface temperature are taken from the level 2 CLIMCAPS

atmosphere cloud and surface geophysical state V2 data product (Barnet, 2019a). Using the cloud cleared radiances allows us to avoid cloud handling in the retrieval process and including cloudy pixels provides more collocated pixels for TROPOMI. CrIS has a field of view consisting of $3 \times 3$ circular pixels of 14 km diameter each (nadir spatial resolution). In conjunction with the cloud clearing algorithm and due to the subsequent L2 processing, the nine field of view pixels are combined, resulting in an effective spatial resolution of $42 \times 42$ km$^2$.

## 110 2.3 Validation data: MLS, ozonesondes and lidars

The Microwave Limb Sounder (MLS) on the NASA's Aura satellite launched in July 2004 provides thermal emission measurements from broad spectral bands near 118, 190, 240, 640 and 2500 GHz by seven microwave receivers. Aura moves in a sunsynchronous orbit with an equatorial crossing time of 13:45 local time. The spatial sampling of MLS is ∼6 km across-track and ∼200 km along-track. For collocations with TROPOMI and CrIS the maximum distance between both is chosen to be

two hours and 100 km. Vertical ozone profiles derived from MLS observations were characterised and validated extensively (Froidevaux et al., 2008; Livesey et al., 2008) and their temporal stability proven (Nair et al., 2012). In the L2 product version





5.0 used here the altitude range is from 12 to 80 km with a vertical resolution varying between 2.5 - 3.5 km (Livesey et al., 2020). Above 18.5 km the precision is estimated to be 4 – 7% with an accuracy of 5 – 10%. From the lower stratosphere downward to the troposphere the precision of the individual profiles decrease up to 5 – 100% (depending on the latitude) with an accuracy of 7 – 10%.

To validate the ozone profile in the lower stratosphere and in the troposphere, in-situ ozonesonde measurements and ground based ozone lidar data are used. The ozonesondes are provided by the World Ozone and Ultraviolet Radiation Data Center (WOUDC) (WOUDC Ozonesonde Monitoring Community et al.) and the Southern Hemisphere Additional Ozonesondes (SHADOZ) (Witte et al., 2017, 2018; Thompson et al., 2017; Sterling et al., 2017). Those measurements have a high vertical resolution of 100 – 150 m and are well validated. The precision is on the order of 5%, and the accuracy 5-10% (Deshler et al., 2008; Johnson, 2002; Smit et al., 2007). Around the tropopause layer in the tropics the uncertainties peak and reach about 15 – 20% (Witte et al., 2018). During the time when TROPOMI data are available, 242 collocated ozonesonde measurements from 30 different sites were found. The collocation criteria are a maximum distance of 100 km and a maximum time difference of 24 h. The exact locations can be found in supplement material (Table S1).

To validate the lower levels of the atmosphere, tropospheric lidars are an excellent option. Unfortunately they are not as widely distributed as ozonesonde and stratospheric lidar sites. For the limited TROPOMI/CrIS dataset, ozone profiles from three different locations are available: Table Mountain Observatory (CA, USA), University of Alabama Huntsville (AL, USA) and Observatoire de Haute Provence (France). The vertical range of the ozone profiles from Huntsville and Observatoire de Haute Provence (OHP) is 3 – 14 km with a precision of better than 10% (Kuang et al., 2013; Gaudel et al., 2015). The tropospheric lidar measurements are done at day-time in Huntsville and after sunset in OHP. For Table Mountain, where ozone profiles at day-time and night-time are available, the vertical range is increased up to 25 km during the night. The overall precision reaches from 5% in the free troposphere to up to 15% above 20 km (Leblanc et al., 2016).

For a comparison of the lower vertically sampled retrievals to the fine sampled tropospheric lidar and ozonesondes, a pseudo-inverse (linear) regridding (Rodgers, 2002, Sec. 10.3.1) from the finer to the coarser grid is performed. Therefore, the interpolation matrix $L$ is inverted to

$$L^* = (L^T L)^{-1} L^T. \tag{1}$$

The pseudo-inverse matrix $L^*$ is applied to the fine lidar grid $x_{fine}$ as follows

$$x_{coarse} = L^* x_{fine}. \tag{2}$$

## 3 Retrieval method

The ozone profiles are retrieved with the IUP Bremen TOPAS (Tikhonov regularized Ozone Profile retrievAl with SCIA-TRAN) algorithm as applied to TROPOMI UV measurements. It is based on the first-order Tikhonov regularisation approach (Tikhonov, 1963) and is described in detail by Mettig et al. (2021).





In general the TOPAS algorithm comprises three steps within each iteration. The first is the radiative transfer model (RTM) calculation, where a radiance spectrum is simulated using the a priori information or the retrieval results from the previous iteration. The second step is a pre-processing to account for effects that can not be handled within the RTM, for instance, the secondary calibration among others. In the final step, the physical quantities contained in the state vector $x$ are determined. At the $i$-th iterative step the solution is given by

$$x_{i+1} = x_a + [K^T S_y^{-1} K + S_r]^{-1} [K^T S_y^{-1} (y - F(x_i) - S_r(x_i - x_a)]. \qquad (3)$$

Here, the forward model simulation $F(x)$ is compared to the measurement vector $y$ while the a priori state vector $x_a$ is compared to the state vector from the last iteration (or first guess values) $x_i$. The Jacobian matrix of the forward model, $K$, is also referred to as the weighting function matrix. The constraints are the measurement error co-variance matrix $S_y$ and the 1-st order Tikhonov regularisation matrix $S_r$. The retrieval step comprises information from both UV and IR spectral ranges. Essential retrieval settings are listed in Table 1.

**Table 1.** Settings of the TOPAS retrieval step

| Parameter | Setting |
|---|---|
| Retrieved quantities | Vertical ozone profile (UV and IR) |
| | Integrated water vapour column (IR) |
| | Scalar albedo (UV) |
| Wavelength range | 270 – 329 nm and 9350 – 9900 nm for ozone profile (UV and IR) |
| | 9350 – 9900 nm for water vapour column (IR) |
| | 310 – 329 nm for scalar albedo (UV) |
| Regularisation | Tikhonov 0th order parameter: 11.11 (corresponds to a priori variance of 30%) |
| | Tikhonov 1st order parameter: 0.02 above 20 km, |
| | linear interpolation between 16 km, 10 km, 6 km and 1 km |
| | with values of 0.06, 0.1, 0.06 and 0.02, respectively |
| Measurement error co-variance entries | Taken from fit residuals obtained from the pre-processing step |
| Vertical grid | 0 - 60 km, 1 km steps |
| Ozone profile climatology | Lamsal et al. (2004) |
| A priori total column ozone | WFDOAS retrieval (Weber et al., 2018) |
| A priori albedo | WFDOAS retrieval (Weber et al., 2018) |
| Temperature and pressure profiles | ECMWF ERA5 reanalysis (Hersbach et al., 2020) |
| Convergence criteria | 2% change of the ozone profile or the spectral fit RMS |

Compared to the ozone profile retrieval from TROPOMI UV data described in Mettig et al. (2021), there are two main changes: the way the measurement error co-variance matrix $S_y$ and an altitude-dependent 1st order Tikhonov regularisation is constructed (see Table 1). For $S_y$, the fit residuals from the pre-processing step are used instead of instrument SNRs. Tests have shown that this approach works better for combined UV+IR retrievals. The underlying problem is the different SNR of

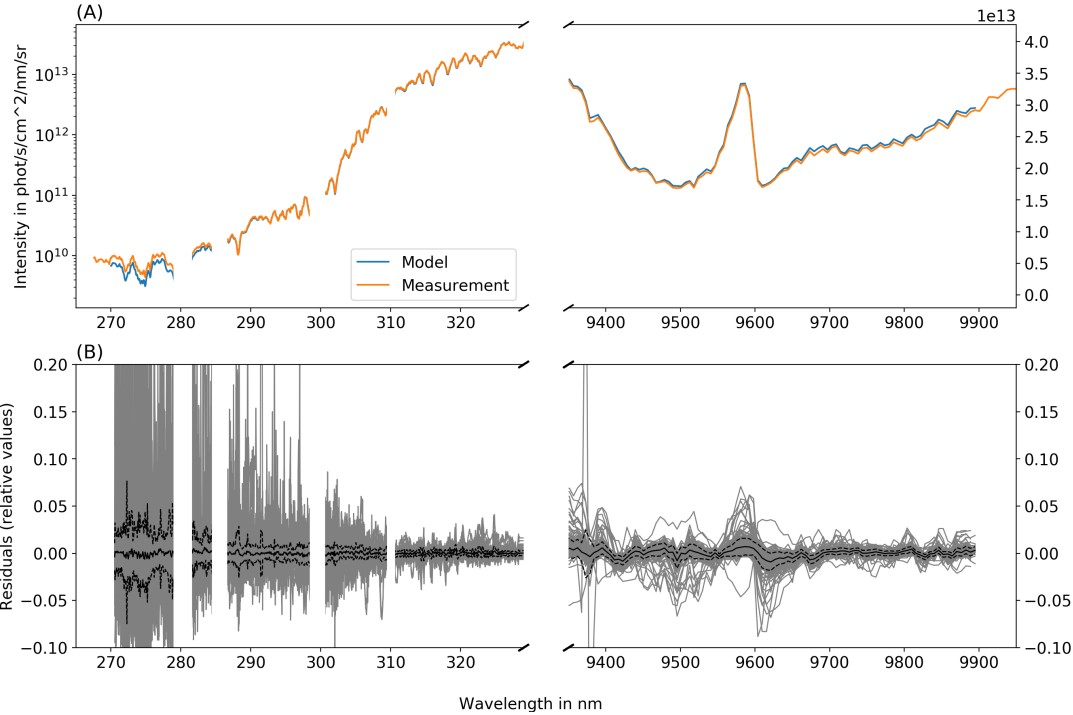

**Figure 1.** (A) Measured intensity from TROPOMI (270 – 329 nm) and CrIS (9350 – 9900 nm) on 6 July 2018 at 12:11 p.m. (TROPOMI) compared to the modelled intensity after the iterative process. (B) Residuals from logarithmic fits for 242 ozone profile retrievals where collocated ozonesonde measurements are available. The residuals of individual retrievals are plotted in grey while the mean of all residual spectra (solid line) and its standard deviation (dashed line) is given in black.

TROPOMI and CrIS and the different spatial resolution of the instrument's measurements from the binning of the pixels. The huge and fluctuating differences between the SNR in UV and IR make it nearly impossible to stabilize the retrieval for all

possible conditions. The use of fit residuals as measurement error co-variance for both spectral ranges mitigates this problem and enables retrievals with constant settings which deliver meaningful results under all measurement conditions. The 1st order Tikhonov regularisation parameter is no longer constant but is now altitude dependent. With the inclusion of the IR spectral range, the information content in the troposphere increases. To make optimal use of this fact, the regularisation below 20 km is weakened. Above 20 km, the Tikhonov parameter is constant and is 0.02. Below, the values are linearly interpolated between

the altitudes 16, 10, 6, and 1 km. Values are: 0.06, 0.1, 0.06 and 0.02, respectively.

The simulated UV and IR intensities from which the residuals are calculated are shown in Fig. 1. Spectral points are cropped out at the spectral window limits in the region of 300 and 310 nm and at the wavelengths of 280 nm and 283 nm, which contain the magnesium Frauenhofer lines. A comparison between modelled and measured intensity in the UV and IR spectral ranges (A) and residuals for 242 ozone profile retrievals collocated with ozonesonde measurements (B) are shown. In the UV spectral





range the residuals increase for shorter wavelength. While for longer UV wavelengths (310 – 330 nm) the standard deviation
is about 0.25% and increases to of 7.5% at the shortest UV wavelengths.

## 3.1 UV RTM and preprocessing

In the UV spectral range, the RTM simulates TROPOMI measurements assuming a pseudo-spherical atmosphere with the
ozone absorption cross-sections from Serdyuchenko et al. (2014) convolved with the TROPOMI instrument response func-
tion (ISRF) (ESA/KNMI, 2021). Other input parameters in the forward simulations are: the measured solar spectrum from
TROPOMI, the viewing geometry angles, the effective scene height as well as a piori values for ozone (profile and total col-
umn amount) and albedo. Temperature and pressure profiles are taken from ECMWF ERA-5 reanalysis. The polarisation and
the rotational Raman scattering, which have a significant impact in the UV spectral range, are ignored in the RTM for computa-
tional reasons. They are accounted for in the pre-processing step using lookup-tables (LUT). The polarisation is described by a
wavelength dependent factor applied to the measured spectra, which is given by the ratio of polarised and unpolarised synthetic
intensities calculated for appropriate values of the viewing geometry angles, albedo, total ozone and scene height. Another part
of the pre-processing is the subtraction of a polynomial, spectral fitting of three pseudo-absorbers, and wavelength adjustments
(shift and squeeze correction). In total three different pseudo-absorber parameters are fitted:

- the rotational Raman scattering (Ring) correction, which is given by a LUT in the same manner as the polarisation
correction (ratio of spectra with and without Raman effect),

- the re-calibration spectrum, which is determined by a comparison of TROPOMI measurements with simulations using
    MLS ozone profiles,

- the inverse solar irradiance spectrum representing a wavelength independent offset in the measured data

The pseudo-absorber fit and the shift and squeeze correction are performed for each of the separate UV spectral windows listed
in Table 2 independently. A linear polynomial is subtracted in the lower UV2 spectral window only, while no polynomials are
subtracted in the other UV spectral windows.

## 3.2 IR RTM and preprocessing

In the IR wavelength range, the intensities are simulated using a line-by-line model, which is also part of SCIATRAN-V4.5. The
spectroscopy database HITRAN (HIgh-resolution TRANsmission molecular absorption database) 2020 (Gordon et al., 2021)
is used. A continuous spectrum between 9350 and 9900 nm with a sampling of 0.05 nm is modelled containing atmospheric
trace gases $O_3$, $H_2O$, and $CO_2$ in the forward model. Because $CO_2$ does not affect the ozone profile retrieval, it is kept
constant using a climatological $CO_2$ profile calculated with B2D chemistry-transport model (Sinnhuber, 2003). The change
of water vapour is taken into account by retrieving the integrated column value and scaling the climatological $H_2O$ profile.
The rotational Raman scattering and polarisation are not taken into account as the contribution of scattered solar radiation is
negligible. The surface emissivity is set to unity and is not changed during retrieval. Instead, the contribution of the surface





**Table 2.** TOPAS RTM settings and pre-processing steps in the UV.

| Parameter | Setting |
|---|---|
| Radiative transfer model | SCIATRAN V4.5 |
| | pseudo-spherical atmosphere |
| | no polarisation and no rotational Raman scattering |
| Ozone absorption cross-section | Serdyuchenko et al. (2014) |
| TROPOMI spectral resolution and sampling | 0.5 nm resolution, 0.065 nm sampling |
| Spectral windows | UV1: 270 – 300 nm |
| | lower UV2: 300 – 310 nm |
| | upper UV2: 310 – 329 nm |
| Cloud handling | Effective scene height (cloud fraction weighted mean of surface altitude and cloud height) |
| Polarisation correction | multiplicative spectral correction given by a LUT |
| Pseudo-absorbers | Ring correction: multiplicative spectral correction given by a LUT |
| | Re-calibration: correction spectra resulting from simulations using MLS ozone profiles for selected orbits |
| | Offset correction: inverse solar irradiance spectrum |
| Polynomial | linear polynomial in the lower UV2 (300 – 310 nm) channel |

emission is approximated by a polynomial and subtracted from the measured and modelled spectra within the pre-processing step. The surface temperature is taken from the CrIS L2 product (Barnet, 2019a). Temperature and pressure profiles are taken from the ECMWF ERA-5 reanalysis data, same as for the UV range. Typical spectral residuals, which are used to initialise the error co-variance matrix, are shown in Fig. 1. In the IR spectral range the residuals scatter in the 1% range. In comparison,

the noise measured by CrIS is 10 to 20 times smaller. The use of higher noise levels as weights in the retrieval reduces the information content of our retrieval. If the original CrIS noise is used in an IR-only retrieval, the DOF increases from 2 – 2.5 to 3 – 3.5. In the combined retrieval, however, we have to make a compromise in order to guarantee stability using both UV and IR wavelength ranges as mentioned above.

During the pre-processing step, the modelled radiance is convolved with the spectral response function of the CrIS instru-
ment, which is represented by the Hamming function (Han et al., 2015). A linear polynomial is included in the fit to account for the surface emissivity. The settings for RTM and the pre-processing in the IR are listed in Table 3.

## 4    Retrieval characterisation and comparison: UV, IR, and UV+IR retrievals

Our approach to compare the combined UV+IR ozone profile retrieval with UV-only and IR-only retrievals is based on the principle that all retrievals should be as similar as possible. All retrievals are run using the settings optimised for the combined





**Table 3.** IR settings of the TOPAS RTM and pre-processing

| Parameter | Setting |
|---|---|
| Radiative transfer model | SCIATRAN V4.5 line-by-line |
| Molecular spectroscopic database | HITRAN 2020 (Gordon et al., 2021) |
| CrIS spectral resolution | $0.625\ \mathrm{cm}^{-1}$ (equivalent to $\sim 5$ nm) |
| Spectral windows | 9350 – 9900 nm, 0.05 nm steps |
| Convolution | Hamming function (Han et al., 2015) |
| Emissivity | Set to unity |
| Water vapour a priori profile | Standard atmosphere |
| Surface Temperature | CrIS L2 surface temperature product (Barnet, 2019a) |
| Polynomial | linear polynomial |

retrieval with corresponding spectral ranges switched off for the UV-only and IR-only retrievals, respectively. This approach represents the most straightforward way to analyse the impact of combining both spectral ranges.

Figure 2 shows as an example comparisons of results at a single location from our UV+IR, UV-only, and IR-only retrievals with collocated ozone profiles from tropospheric and stratospheric lidars, ozonesonde, and MLS. All measurements were performed within a 100 km radius around the Table Mountain Facility (34.4° N, 117.7° W) on 27 September 2018. The various
collocated profiles from the satellites and ground are shown in panel (A) with a focus on the stratosphere and (C) with a focus on the troposphere. Corresponding relative differences between profiles are shown in panels (B) and (D). The a priori ozone profile, which is shown in grey, contains more ozone below 24 km and less ozone above 24 km in comparison with the lidars, ozonesonde, and MLS. Our three ozone profile retrievals differ in the altitude ranges between 8 and 28 km. Between 10 and 15 km, the UV-only retrieval remains close to the a priori profile. In the troposphere the advantage of UV+IR and IR-only
retrievals over the UV-only is evident. Between 10 and 15 km altitude, differences of more than 100% are observed in the UV-only retrieval, which are reduced by more than 50% in the combined and IR-only retrievals. In the stratosphere, between 20 and 30 km, the comparison between the TOPAS retrievals and the stratospheric lidar and MLS is not as clear as in the troposphere. Here, the combined retrieval agrees with MLS better than the UV-only retrieval, but UV-only agrees better with the lidar. The IR-only retrieval has a slightly positive bias compared to both MLS and lidar. The vertical resolutions of the
three retrievals, which are given by the inverted main diagonal of the averaging kernel (AK) matrix, are shown in panel (E). As is known from previous studies, the UV-only retrieval from TROPOMI measurements (blue) has high vertical resolution above 20 km and reduced vertical resolution between 10 and 15 km (Mettig et al., 2021). The IR-only retrieval from CrIS measurements (orange) has a vertical resolution of around 10 km between 5 and 25 km. The combined UV+IR ozone profile retrieval shows a vertical resolution of about 10 km from 5 to 55 km. The contribution from the IR to the combined retrieval
diminishes above about 30 km meaning that the upper stratosphere is derived mostly from the UV part of the retrieval.

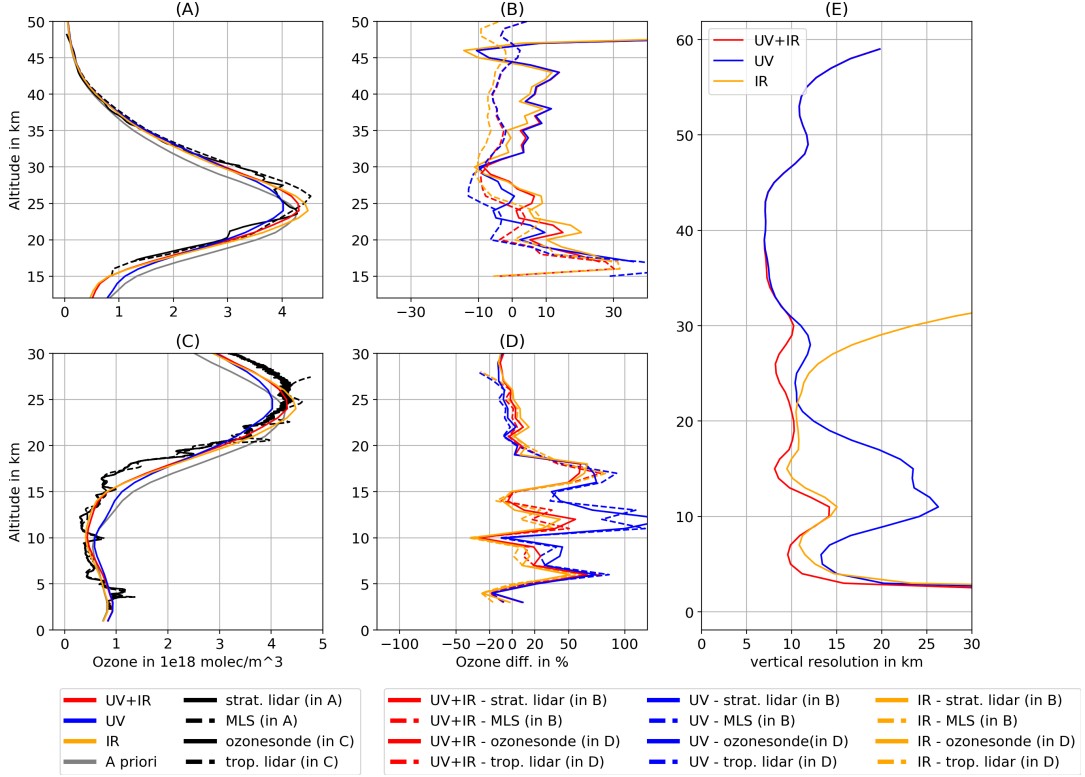

**Figure 2.** Comparison between UV+IR, UV-only, and IR-only ozone profile retrievals and collocated stratospheric lidar and MLS measurements (A and B) and collocated tropospheric lidar and ozonesonde profiles (C and D). The TROPOMI and CrIS data were obtained at 34.2° latitude and -117.9° longitude on 27 September 2018 at 21:20:11. All measurements, the MLS ozone profile, the ozonesonde and tropospheric/stratospheric lidar sounding from Table Mountain Facility, are within 100 km distance of TROPOMI/CrIS. The time difference for is 25 minutes for MLS and between 6 and 8 hours (nighttime profile) for the lidars and ozonesonde. The vertical resolution of the 3 TOPAS retrievals is given in (E).

The distribution of information content in the ozone profile retrievals is presented in more detail in Figure 3, where the rows of the AK matrices are shown. The lack of information between 7 and 17 km in the UV-only retrieval (B) appears partially compensated by the IR retrieval component from the CrIS measurements. Above 30 km, the AKs of UV+IR and UV-only retrieval are the same (diminishing role of IR part). The UV+IR retrieval is, however, not a simple linear combination of the UV-only and IR-only retrievals. This is evident, for example, from the mid-blue contour lines (10 – 15 km). In the UV+IR
retrieval, they display a significant negative peak around 25 km, which is not present in the other two retrievals. It means that the overlapping sensitivity can change the altitude distribution of the information content. Overall, the information content of the UV+IR retrieval increases in contrast to the UV-only and IR-only retrieval, as seen from the DOFs (see text in the panels of Fig. 3). Compared to the UV-only retrieval, DOF increases by almost one for UV+IR. About half of this enhancement comes
from the troposphere and the other half from the lower stratosphere. The tropospheric DOF for the IR-only retrieval is lower

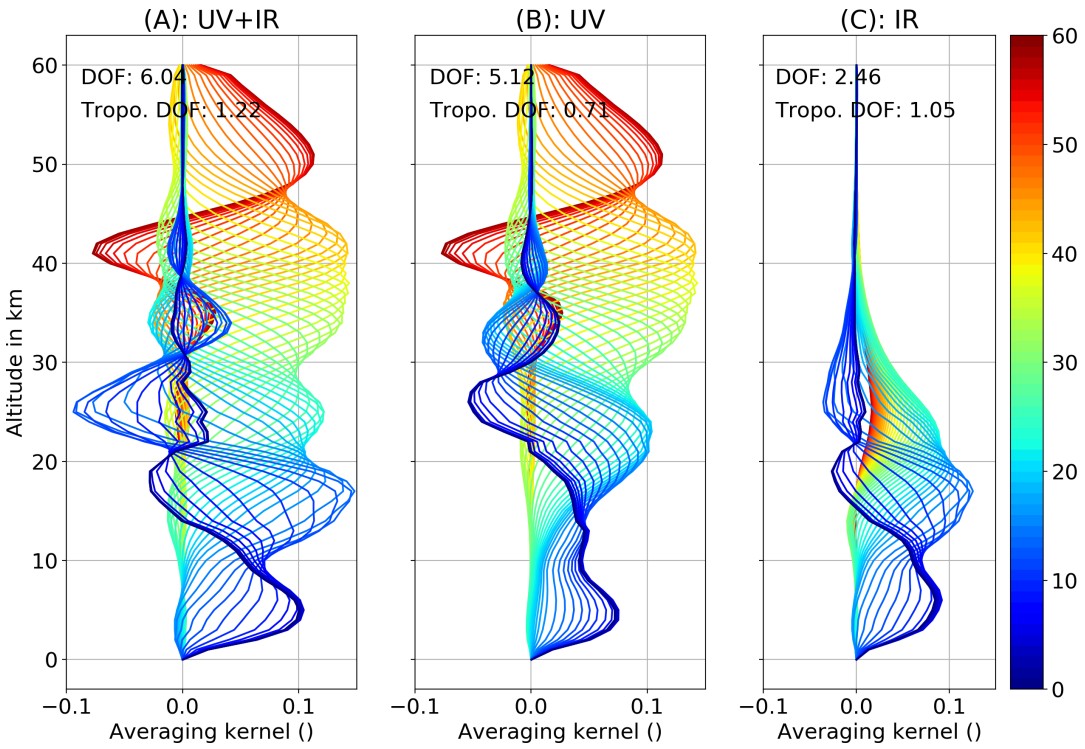

**Figure 3.** Rows of the averaging kernel matrix for the three retrievals shown in Fig. 2. Each line represents the sensitivity of the ozone profile retrieval at a certain altitude level from 0 km (blue) to 60 km (red). DOF is given by the sum of the AK matrix main diagonal. The tropopause is defined by 2 PVU from ERA5 reanalysis.

compared to the values for other IR sensors reported in previous publications (e.g. Cuesta et al., 2013; Fu et al., 2013). This may be due to the lower spectral resolution of CrIS compared to IASI and TES.

## 5 Validation

The validation of the TOPAS UV+IR retrieval focuses here on the troposphere, which we try to improve using the combined
UV and IR retrieval. Profiles and tropospheric ozone content (TOC) resulting from the TOPAS retrieval are compared with measurements from ozonesondes and tropospheric lidars. In the stratosphere, the ozone profiles of the combined retrievals largely agree with those from the UV-only retrievals as shown in Sec. 4; the latter has been validated in Mettig et al. (2021). We only provide some example results in the lower stratosphere.

### 5.1 Tropospheric lidar

For the validation in the troposphere, tropospheric lidar measurements are particularly suitable. There are only three locations where lidar measurements are carried out regularly with a high temporal frequency (up to two times a day) and with which





collocation was found in the TROPOMI test data set period. Since lidars have a high vertical resolution (below 100 m), similar to the ozonesondes, the lidar altitude grid is adjusted in accordance with Eq. (2) before comparisons are made.

Figure 4 shows the comparison of the TOPAS retrieved ozone profiles and tropospheric lidar measurements at three different sites. While the measurements in Huntsville take place during daytime, the ozone profiles in OHP are measured after sunset. Only Table Mountain provides night- and daytime measurements where the latter match in time with TROPOMI/CrIS overpasses. Nighttime profiles can reach a height of up to 28 km and are used for comparison up to 25 km into the stratosphere. For each of the stations and each retrieval type, the mean ozone profile in number density, the relative mean difference profile in percent, the standard deviation in percent, and the TOPAS vertical resolution are shown. The AK matrix can be applied to
the re-gridded lidar profiles to account for the higher vertical resolution of the lidar measurements. The comparison to lidar profiles convolved with AKs is shown in red, but the results should be taken with care as in the altitude ranges where the combined retrieval is sensitive and a single retrieval is not, the former might appear to be worse. This is because the difference between retrieval and the reference profile multiplied by the AK matrix by definition approaches zero in altitude ranges where the retrieval has low sensitivity, i.e. AKs are close to zero.

At the OHP site (top row of Fig. 4) all retrievals agree well with a relative mean difference within ±10% up to 10 km. Above this altitude, the IR-only retrieval shows better results, but the accuracy of the lidar data decreases here. For Huntsville, where we have the lowest number of collocated profiles, the best agreement with the lidar is found for the combined UV+IR retrieval below 10 km. UV-only and IR-only show a negative bias up to -20%. Above 10 km, the UV+IR and UV-only retrievals are both within the ±10% range. At the Table Mountain site, a clear improvement is seen for the UV+IR retrieval. For both daytime and
nighttime, the combined retrieval shows smaller relative mean differences with respect to the lidar measurements. The IR-only retrieval is slightly better than the combined retrieval in the 10 – 15 km range but has a negative bias up to -20% below 10 km. Between 8 and 18 km the UV-only retrieval remains close to the climatology, while the combined and IR-only retrievals are closer to the lidar measurement. The standard deviations for all comparisons are similar to those of the a priori profiles. The differences between the three retrievals are not that large, but the standard deviations of the UV+IR and IR-only retrievals tend
to be smaller than that of the UV-only retrieval. For the vertical resolution, the conclusion from Figure 2 (C) that the vertical resolution of the UV+IR retrieval is typically better than that of the single retrievals is confirmed. In Huntsville the vertical resolution below 10 km is worse in comparison to the combined and UV-only retrieval. The reason might be the higher viewing angle of most of the collocated CrIS measurements (compared to the other stations).

The absolute difference in the tropospheric ozone content (TOC) for each site is shown in Fig. 5. To obtain these results, the
ozone profiles are integrated from the lowermost retrieval layer altitude up to the tropopause. The tropopause height is obtained from the ERA5 reanalysis data set using the 2 PVU definition. For cloudy pixels, the lidar profile is cut at the effective scene height. Overall good agreement with the lidar TOC is found for the UV+IR retrieval and an improvement in comparison to the UV-only retrieval is observed. For OHP, the UV-only retrieval already has a very small bias of -0.21 DU. The UV+IR retrieval also has a small bias (-0.83 DU) and in addition, the standard deviation is reduced by nearly 1 DU. For Huntsville, the a priori
TOC does not statistically agree with the lidar TOC. Again, the UV+IR and UV-only retrievals can significantly improve the agreement and reduce the TOC difference from -5.64 DU to -3.1 DU and -3.66 DU, respectively. The combined retrieval shows





**Figure 4.** Ozone profile comparison between UV+IR, UV-only, and IR-only retrievals and tropospheric ozone lidar measurements from three different sites. Table Mountain provides in addition daytime (matching S5P/CrIS overpasses) profiles. Nighttime profiles can reach a height of up to 28 km and are used for comparison up to 25 km. In the panel showing the difference, the grey shaded area marks the ±10% range.





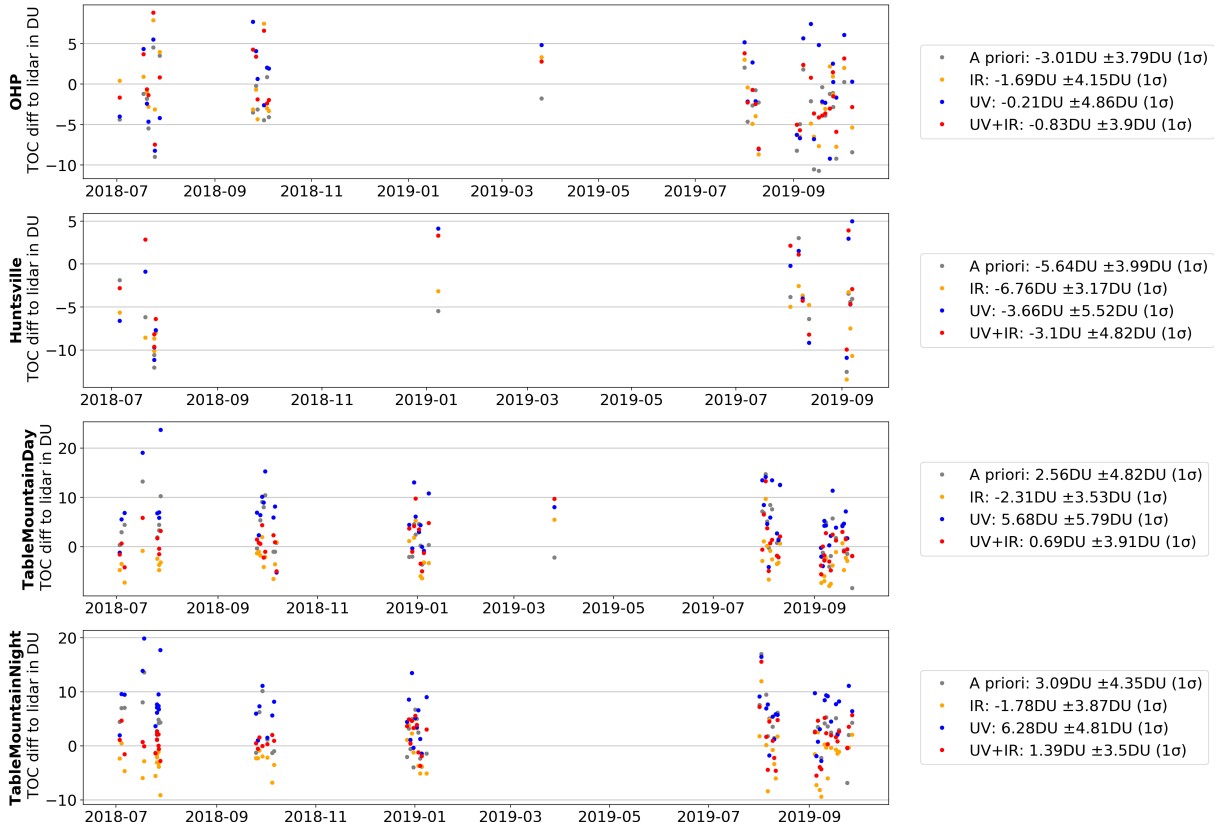

**Figure 5.** Absolute differences in the tropospheric ozone content (TOC) with respect to the tropospheric lidar data. TOCs are calculated by integrating ozone profiles from the lowermost retrieval level altitude up to the tropopause. The height of the tropopause is obtained from the ERA5 reanalysis data using the 2 PV definition. In the legend the mean absolute differences and the standard deviations are given.

a smaller standard deviation in comparison to UV-only. For Table Mountain (day and night) the UV+IR retrieval shows the best results. TOC from the combined retrieval has the smallest bias (0.69 DU for daytime and 1.39 DU for nighttime measurements) and the lowest standard deviation in comparison to single retrievals and to the a priori. For all sites we found that TOC from the IR-only retrieval has a negative bias with a relatively small standard deviation.

Figure 6 shows the TOC comparison to Table Mountain daytime measurements as a scatter plot. One notices that the linear regression line for the UV+IR retrieval (red) agrees very well with the one-to-one line (back dashed). The UV-only retrieval overestimates the TOC, while the IR-only retrieval underestimates it. The correlation between TOCs from the TOPAS retrievals and lidar data is quantified by the R-values, which do not differ much from each other. Only for the UV-only retrieval the correlation is below 0.8. The results for the other stations are given in the supplement (Fig. S1).

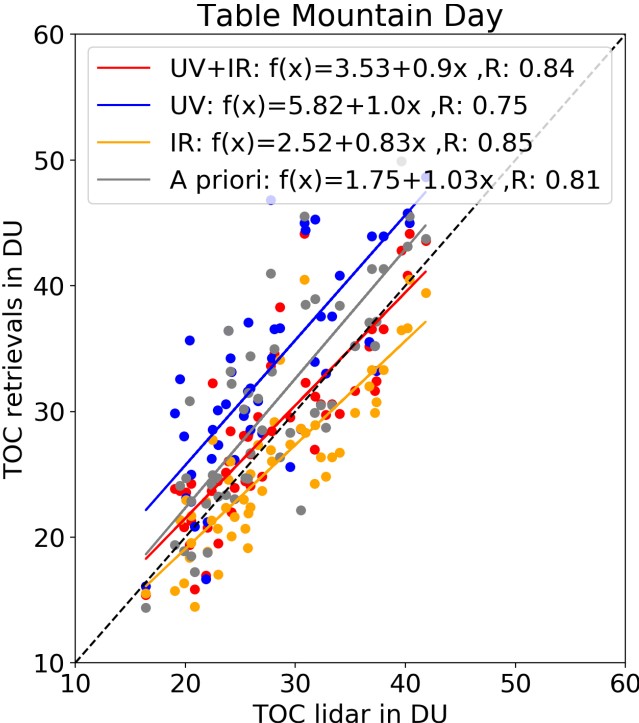

**Figure 6.** TOC scatter plot of TOPAS retrievals with respect to Table Mountain tropospheric lidar data (daylight measurements). The one-to-one line is given by the black dashed line. The linear regression curves are plotted with different colours and their equations are given in the legend.

### 5.2 Troposphere and lower stratosphere: ozonesondes

Overall we found 205 globally distributed ozone soundings, which are collocated with TROPOMI and CrIS. Figure 7 shows a comparison of the three different ozone profile retrievals with ozonesonde data in the tropical region (-20° to 20°) and northern mid-latitudes (20° to 60°). Much fewer collocated data were available in other latitude regions making the comparisons less
reliable. They are shown in the supplement, see Fig S2. The overall findings are similar to those for the tropics and northern latitudes, as discussed below.

Looking at the mean ozone profiles in both latitude regions shown in Fig. 7 it is apparent that the UV-only retrieval results already agree very well with the ozonesonde profiles. Potential improvements from using a combined retrieval can therefore be only minor. In the tropics, the UV+IR retrieval shows good agreement below 7 km, a positive bias of about +20% between
10 – 15 km, about 10% positive bias between 15 and 22 km, and is within ±10% range above 22 km. The UV-only retrieval has a slightly positive bias of about +15% below 10 km but it agrees very well with ozonesondes above 10 km. The IR-only retrieval agrees well with the ozonesondes below 15 km but has a positive bias in the stratosphere. The combined retrieval shows traceable impacts from both independent retrievals but it does not seem to improve results everywhere. The altitude

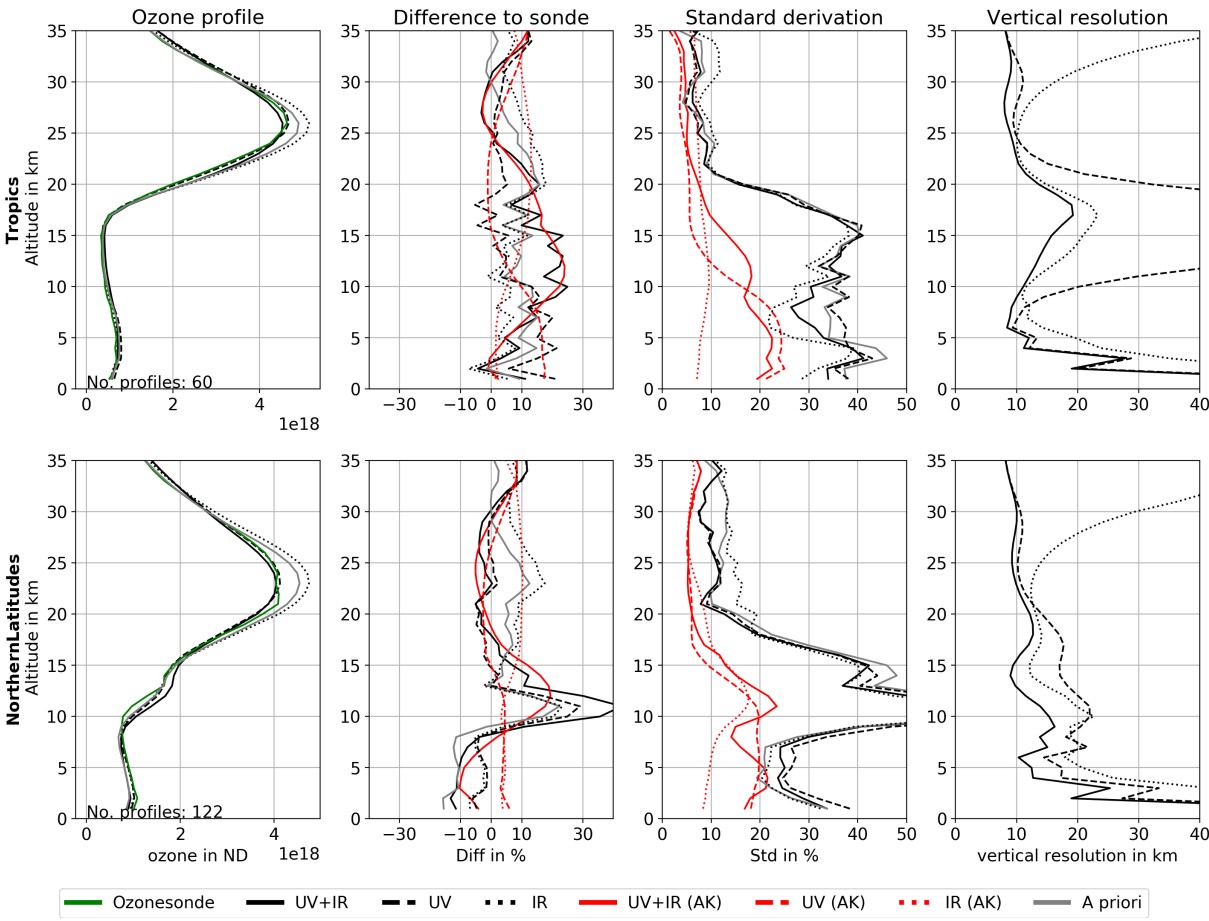

**Figure 7.** Ozone profile comparisons between UV+IR, UV-only and IR-only retrievals and ozonesonde measurements for northern and tropical latitudes.

region between 10 and 15 km is quite challenging in general. This is because the ozone values are lowest in this altitude
range approaching the detection limits of the ECC sensors used in ozonesondes. From Witte et al. (2018), it is known that
ozonesondes in the tropics have an uncertainty up to 15% in the vicinity of the tropopause. At northern latitudes the results
show similarities to the tropics. The UV+IR and UV-only retrievals agree very well with the ozonesonde profiles above 15 km
in the stratosphere. The IR-only retrieval has a positive stratospheric bias similar to the tropics. Near the tropopause, the UV+IR
retrieval shows large positive differences (more than 40%), while the UV-only and IR-only profiles stay close to the a priori
with a +20% bias. Below 7 km, the UV-only and IR-only retrievals agree well with the ozonesondes while the combined
retrieval shows a slight negative bias of -10%. The standard deviations are comparable to those obtained in the comparisons
with the tropospheric lidar data (Fig. 4). The vertical resolution shows a strong dependence on latitude. This dependence is
due to the different solar zenith angles and the typically low ozone content in the tropical UTLS region. Vertical resolution
improves if the light path gets larger (high solar zenith angle) and high ozone at a given altitude (more ozone the better).

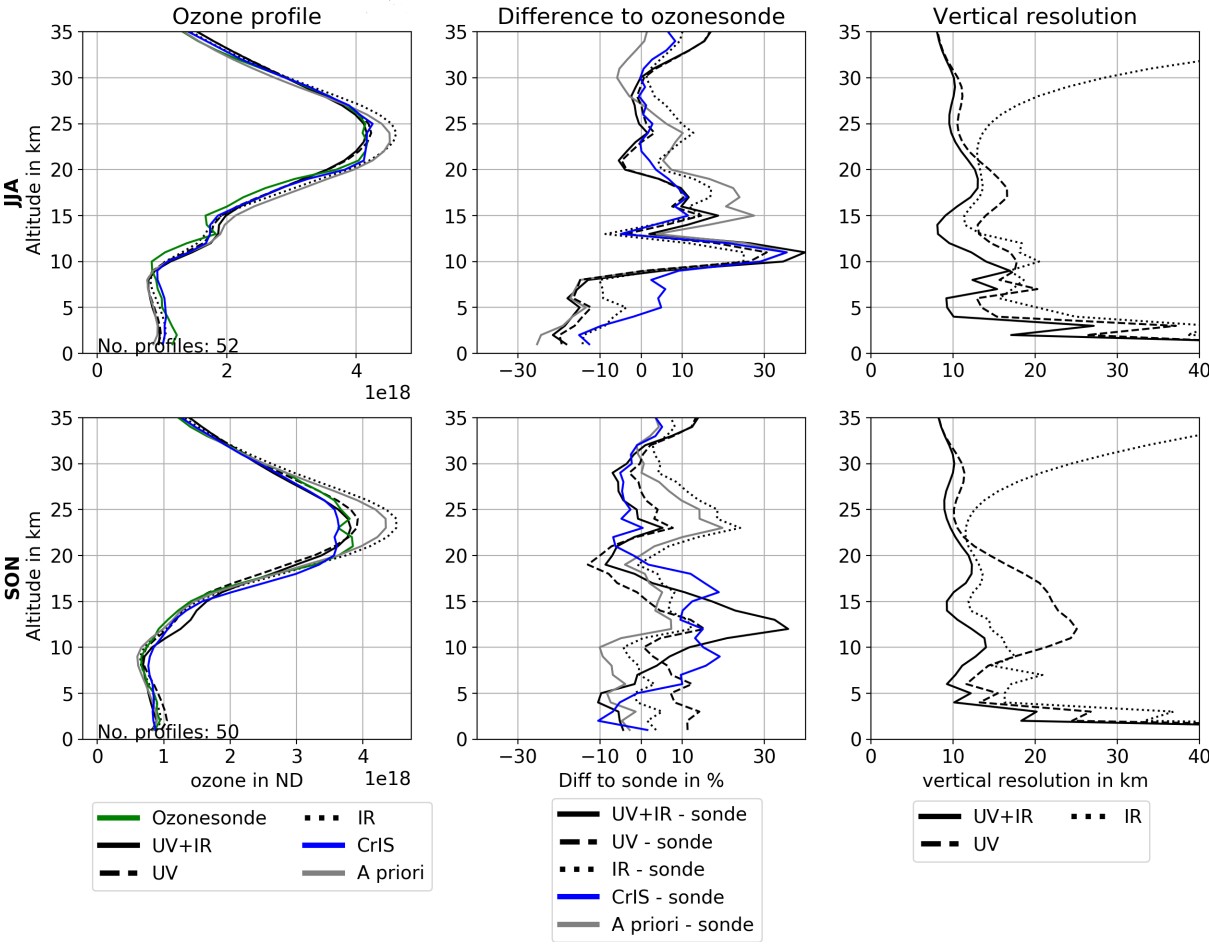

**Figure 8.** Ozone profile comparisons using ozonesondes and NASA operational CrIS profiles at northern latitudes (20°N – 60°N) separated into summer (JJA) and autumn (SON) seasons. The differences to MLS and CrIS data are given only for the UV+IR (solid line) and UV-only retrievals (dashed line).



To investigate the larger differences observed from the combined retrieval, the validation results for northern latitudes are separated into seasons. Figure 8 presents the results in summer (JJA) and autumn (SON) (both seasons with most collocations) while plots for other seasons are provided in the supplement (Fig. S3). The mean collocated ozone profiles from NASA's operational CrIS level 2 product for the same ozonesonde measurements are also shown. For the comparison with CrIS, it must be taken into account that the NASA operational retrieval provides only about 2 DOFs. High vertical sampling of the

CrIS data and its good accuracy in the stratosphere and troposphere results to some extent from the use of the MERRA2 ozone profile data as a priori information (Wargan et al., 2017). In the comparison with the ozonesondes (black solid line), a positive bias of up to 40% is found for the combined retrieval in the altitude region between 10 and 15 km in both seasons, as mentioned above. However, there are two different situations to be considered. In summer, the a priori profile does not agree well with the ozonesonde data and none of the retrievals can substantially improve it. The results from the three retrievals

are very similar. The UV-only retrieval already has a quite good vertical resolution under the given conditions and no further improvement can be achieved by adding the IR measurements. CrIS/MERRA2 ozone profiles (blue) shows the same differences to the ozonesondes as the TOPAS retrievals, which means they agree very well. It should be noted that Wargan et al. (2017) also reported deviations between MERRA2 and ozonesondes of up to ±30% in the tropopause region for 2003 and 2005. In autumn, the situation is different. The mean a priori profile agrees well with the ozonesondes. Because the UV-only retrieval

has a low vertical resolution between 10 and 15 km, it remains close to the climatology. Below 10 km it shows a slight positive deviation of +10%. In contrast, the UV+IR retrieval shows a positive difference of +35% at 12 km and a 10% negative bias at 5 km. The CrIS product has a similar shape of the difference profile. The positive bias peak between 8 and 18 km seems to be smoothed and less pronounced, but still exists.

     The reason for the positive bias between 10 – 15 km might be a compensation effect that occurs when both spectral ranges

are combined. It is known from previous studies (Boynard et al., 2016; Dufour et al., 2012; Nassar et al., 2008; Worden et al., 2007b; Verstraeten et al., 2013) that retrieved ozone profiles from nadir-viewing IR instruments show a positive bias in the UTLS and in the stratosphere above (20 – 30 km). Boynard et al. (2016) showed that ozone profiles retrieved from IASI have a clear positive bias of up to +30% in the tropics and +10% in the middle latitudes between 20 and 35 km. In the UTLS region a positive bias up to +40% was found in the tropics and polar regions. Dufour et al. (2012) found similar results in the

UTLS from comparing three different IASI ozone profile algorithms. Ozone profile retrievals using IR measurements from TES (Worden et al., 2007b; Nassar et al., 2008; Verstraeten et al., 2013) point out similar features, an overestimation of ozone of up to +20% in the stratosphere (20 – 30 km) and in some latitude regions in the UTLS as well. The cause of this stratospheric bias is not yet fully understood. Possible explanations include insufficient vertical resolution of the ozone profiles, instrument artefacts, spectroscopy problems, forward model errors, and insufficient quality of a priori profiles (Verstraeten et al., 2013;

Boynard et al., 2016). For the combined retrieval, the retrieval solution in the 20 – 30 km altitude range is dominated by the UV measurements and thus it does not show the typical bias of the IR retrievals. On the other hand, the uncompensated contribution from the IR spectra in this altitude range gets then balanced by over-correcting the profiles at lower altitudes, where the sensitivity of UV measurements is low. From AKs shown in Fig. 3 we see that a positive variation of the true state around 15 km causes a significant negative response around 25 km. Thus, the positive bias around 25 km present in the IR-only





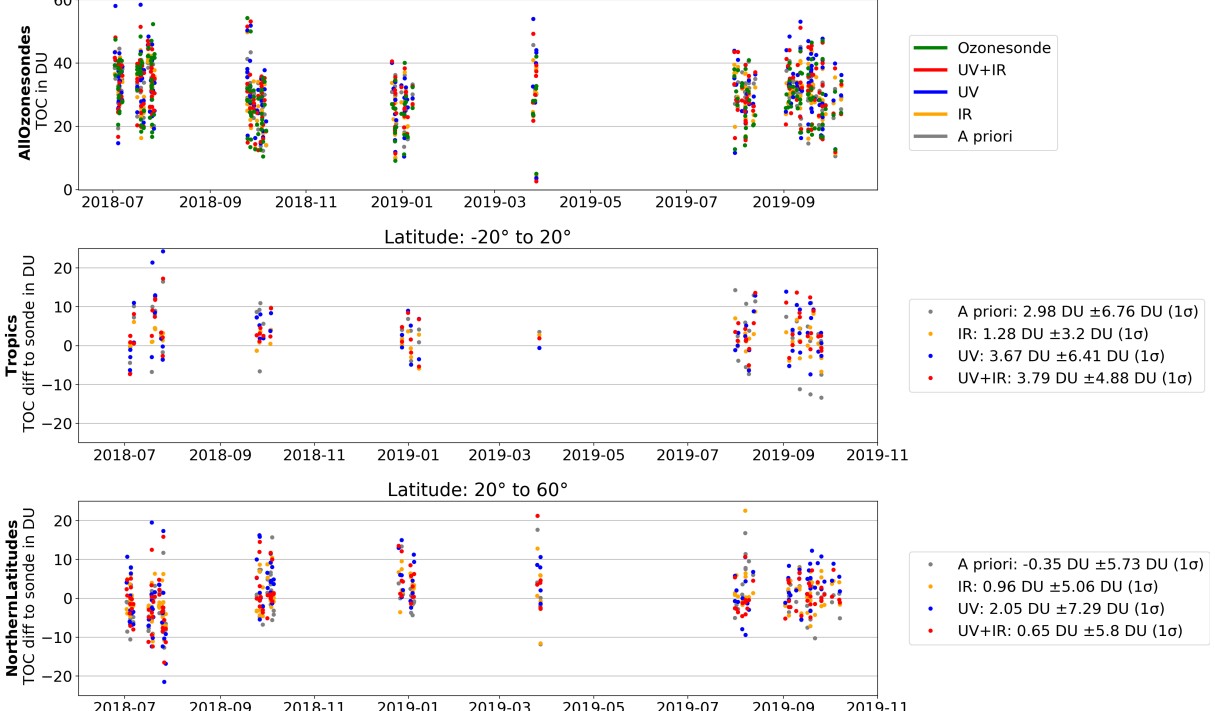

**Figure 9.** Comparison of tropospheric ozone content (TOC). Upper panel: TOC from TOPAS retrievals and ozonesondes. Middle panel: Absolute differences in TOC with respect to the ozonesonde data in the tropics (20°S – 20°N). Lower panel: same as the middle panel but for the northern latitudes (20°N – 60°N). TOCs are calculated by integrating ozone profiles from the lowermost retrieval level altitude up to the tropopause. The height of the tropopause is obtained from the ERA5 reanalysis data using the 2 PVU definition. In the legend (to the right), the mean absolute differences and the standard deviations are given.

retrieval and removed in the combined retrieval, might be compensated in the IR part of the combined retrieval by increasing the values around 15 km. From the comparison with the tropospheric lidar shown in Fig. 2 it is also seen that the UV+IR retrieval performs well around 15 km when the IR-only retrieval does not show a positive bias in the stratosphere. The reason why in this particular case the stratospheric IR-only results are nearly bias-free remains to be investigated.

      In Fig. 9 the TOCs from TOPAS retrievals are compared to collocated ozonesonde data in the tropical region and at northern
mid-latitudes. The comparison results largely confirm the findings from the ozone profile comparisons. The retrievals and the a priori data show a slight positive bias. The best agreement with +1.28 DU ±3.2 DU is found for the IR-only retrieval in the tropics. The combined retrieval has a larger +3.79 DU mean difference but the standard deviation is reduced by 1.5 DU in comparison to the UV-only retrieval and to the a priori. The findings are comparable to the results received from the CCD method using TROPOMI data in the tropics (Hubert et al., 2020). In a validation using SHADOZ ozonesondes, a positive
bias of +2.3 DU with a dispersion ($1\sigma$) of 4.6 DU was shown. At northern latitudes the mean a priori TOC already agrees very well with the mean ozonesonde TOC but the scattering of the results is rather large with a standard deviation of 5.73 DU.





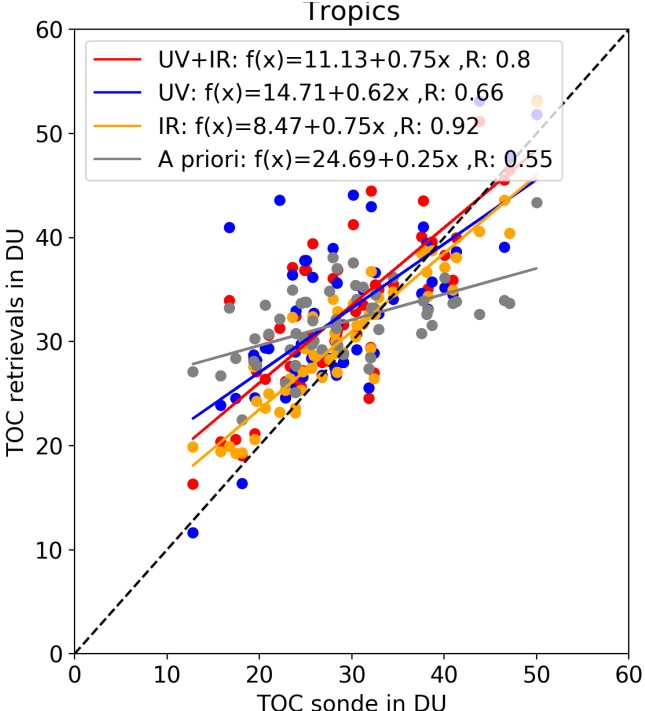

**Figure 10.** Scatter plot of TOCs from TOPAS retrievals with respect to ozonesonde data in the tropics. The one-to-one line is given as a black dashed line. The linear regression curves are plotted with different colours and their equations are given in the legend.

Neither IR-only nor UV+IR retrieval can significantly improve the results here. However, a comparison with UV-only TOC and standard deviation shows that the combined retrieval improves both the TOC and the scattering of differences. Tropospheric ozone retrieved from the combined retrieval is improved compared to the results from the UV-only, even if the UV+IR profile 380 has a larger bias at 12 km.

An additional assessment of the retrieval quality is presented in Fig. 10 as a scatter plot of TOCs from the various TOPAS retrievals with respect to ozonesonde data. This plot shows the results from the tropical region. A similar plot for the northern latitudes is presented in the supplement (Fig. S4). While in the profile and TOC comparisons no issues could be identified for a priori data, it becomes apparent from the scatter plot that climatological TOCs have a quite low correlation with ozonesonde 385 data resulting in a correlation coefficient of only 0.55. The UV-only retrieval correlates slightly better with a correlation coefficient of 0.66. The IR-only and UV+IR results show much better correlation with the TOCs from ozonesondes with R-values of 0.8 and higher. This means that if only the tropospheric ozone content matters, IR-only retrieval is the best choice in the tropics. If, however, the entire atmosphere is to be considered, then UV+IR retrieval is yields better results.





### 5.3 Lower stratosphere: MLS

As follows from Fig. 2, the inclusion of CrIS IR measurements to the ozone profile retrieval has an impact not only on the troposphere but also on the stratosphere. To assess the effect in more detail, the UV+IR and UV-only retrievals are compared with collocated MLS ozone profiles as an exmaple for one day, October 1st, 2018. About 1400 collocated measurements were identified for this day. Results for some other days are presented in the supplement (Fig. S5 – S8). Figure 11 shows the vertical resolution of the UV+IR and UV-only ozone profile retrievals as a function of latitude and altitude. The UV-only retrieval

has high vertical resolution of about 10 km in the stratosphere between 20 and 50 km. Below 20 km its vertical resolution degrades showing strong latitude dependence due to the viewing geometry and the respective ozone content in the atmosphere. As expected, there are no differences between UV+IR and UV-only retrievals above 30 km (retrieval dominated by the UV range). Between 20 and 30 km the vertical resolution of the UV+IR retrieval is typically higher, which is reflected by the wider areas of dark blue color in the respective contour plot. Only in the northern high latitudes almost no change in the vertical

resolution is observed. The greatest improvement of the UV+IR retrieval vertical resolution in comparison to that of UV-only is observed in the altitude range between 10 and 20 km. At higher latitudes, a vertical resolution of 10 km is achieved, similar to that in the stratosphere. In the tropics, the vertical resolution of UV+IR is also significantly higher, but remains at values between 15 and 20 km, i.e. still the vertical resolution is not sufficient to retrieve an independent sub-column layer from this altitude range. In the northern subtropics near 30 - 35 °N there are particularly significant changes. Here the resolution reaches

more than 28 km (red) in the UV-only retrieval but optimises to 10 km (blue) in the combined retrieval. This is consistent with the very good results achieved from the latter retrieval as seen by the comparisons with tropospheric lidar data. Below 10 km, the differences in the vertical resolutions of both retrievals are less pronounced. In the tropics, the vertical resolution below 10 km altitude is already quite good for the UV-only retrieval and only slightly better for the combined retrieval. In the northern and southern higher latitudes, the vertical resolution of the UV+IR retrieval (15-25 km below 10 km altitude) is better than that

of the UV-only retrieval ($\sim$30 km) but again no independent sub-column layer can yet be determined in this altitude range.

The zonal mean differences between MLS and TOPAS (UV+IR, UV-only and a priori) ozone profiles are shown in Fig. 12. The plot is limited to the 16 – 30 km altitude range because MLS provides the most reliable profiles above 16 km, and differences between UV+IR and UV-only retrievals are observed only below 30 km. The differences of the climatological (a priori) ozone profiles to all observations reaches up to +20% and shows an oscillating pattern above 20 km. Near the equator,

there is a large positive difference of over +30%. Below 20 km, some areas with differences higher than 30% are observed. For both UV+IR and UV-only retrievals, the oscillating positive pattern above 20 km is not present any more while the differences in the tropics and in the troposphere are still observed, although less pronounced. In both UV+IR and UV-only retrievals, the negative differences are more dominant. Overall the relative mean differences of the combined retrieval are lower. Around -20° and +20°, negative differences are observed between 16 and 22 km. The differences for the UV+IR retrieval are by about 20%

smaller here. Above 22 km, the UV-only retrievals show positive differences which are not present in the UV+IR retrieval. At northern latitudes below 60°, the overall mean differences are also lower for the combined retrieval. As the information content from UV measurements is clearly dominating in the stratosphere, the improvement in the stratospheric part of the retrieved





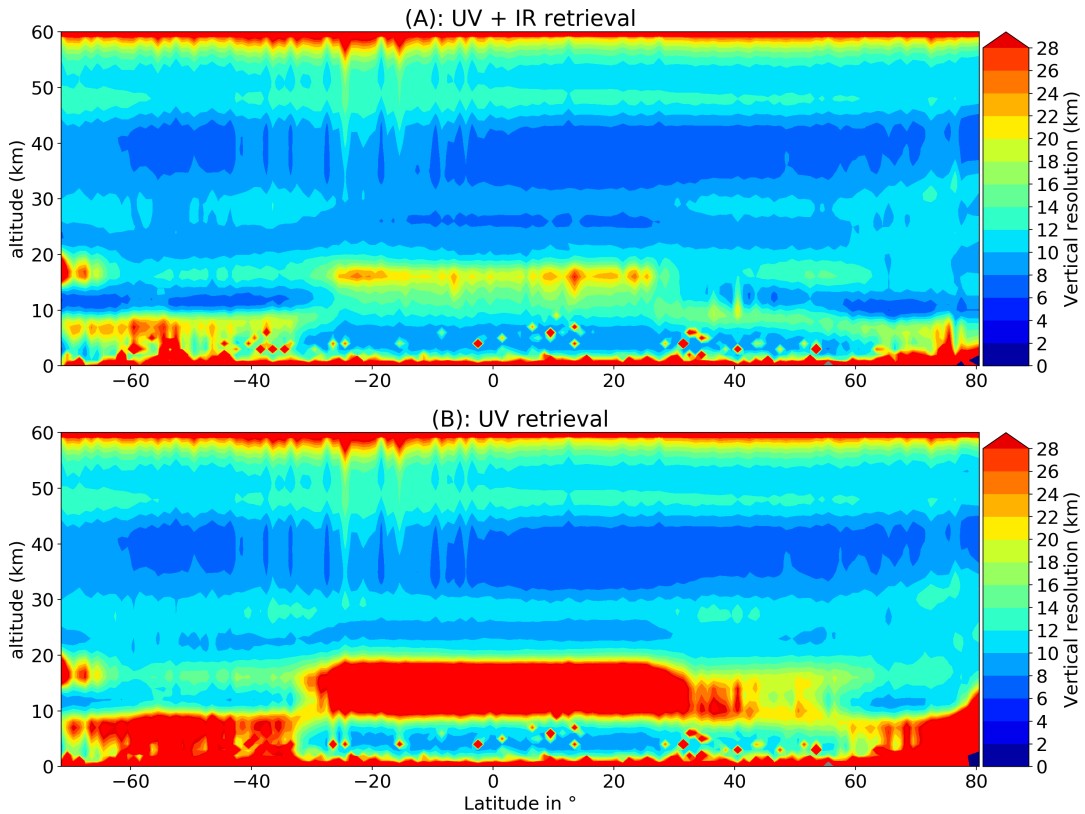

**Figure 11.** Zonally averaged vertical resolution for one day of TOPAS retrieval data (1 October 2018). The vertical resolution is given by the inverse main diagonal of the AK matrix. Upper panel (A): vertical resolution for the combined retrieval. Lower panel (B): vertical resolution from the UV-only retrieval.

ozone profiles due to inclusion of the IR spectral range is rather moderate and not observed on every day. Further studies with a larger amount of data would be helpful to investigate this in more detail.

## 6 Conclusions

Spectral measurements from the instruments TROPOMI and CrIS were combined to improve the ozone profile retrieval using either instrument alone. The combined retrieval is particularly suited for CrIS and TROPOMI, as they fly in the same orbit just a few minutes apart. The combined UV and IR retrieval was successfully implemented by applying our TOPAS algorithm to the UV spectral range of 270 – 329 nm (TROPOMI) and the IR spectral range between 9350 – 9900 nm (CrIS). Advantages of the combined UV+IR ozone profile retrieval were demonstrated by comparing with our UV-only and IR-only retrievals. All TOPAS retrievals were run using the same settings and the same measurement data set. Even though the available TROPOMI





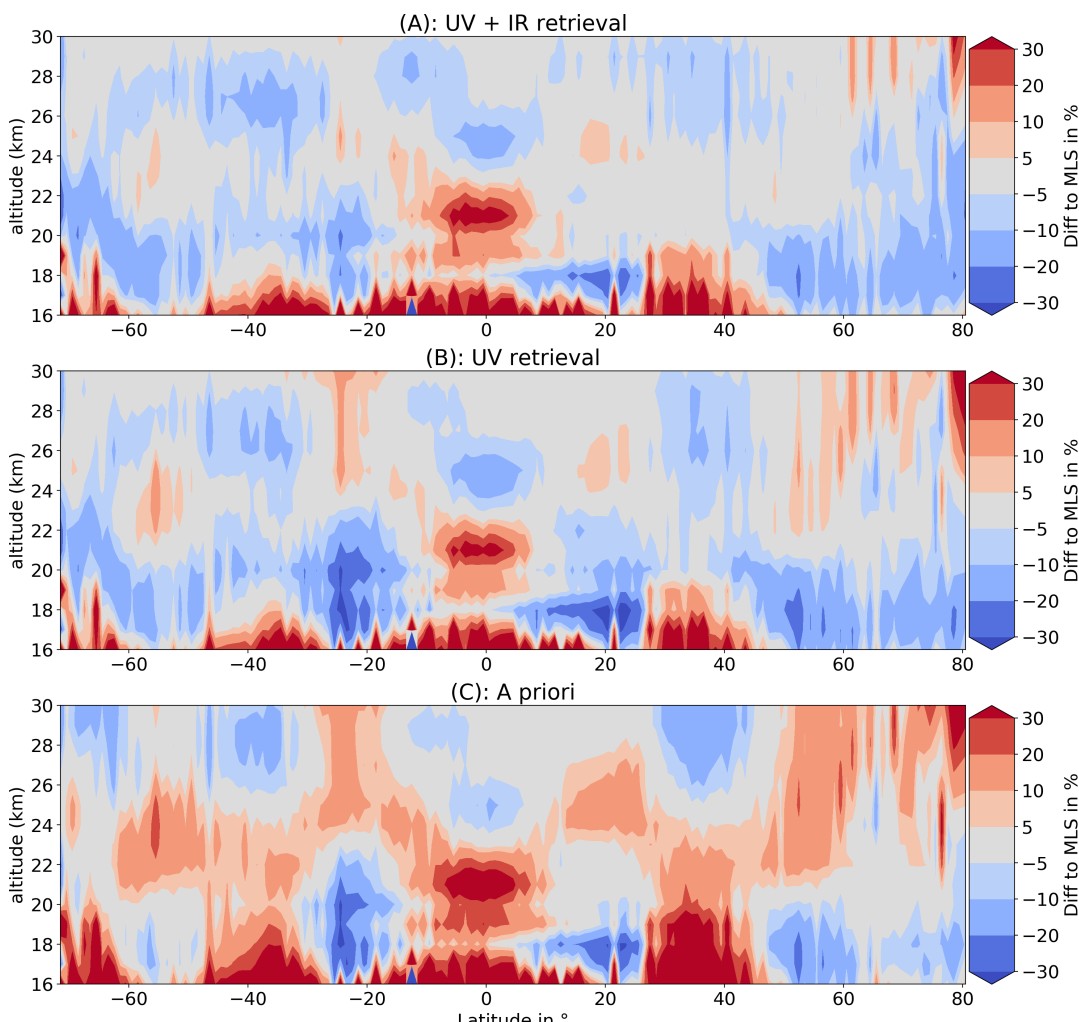

**Figure 12.** Zonal mean differences in percent between combined TOPAS ozone profiles (A), UV-only TOPAS profiles (B) as well as the climatological (a priori) (C) and MLS data on 1 October 2018.





dataset is still very limited, improvements in the UV+IR retrieval were demonstrated by validation with collocated tropospheric lidar, ozonesondes, and MLS data. The main findings are as follows:

– The vertical resolution improves by adding CrIS IR spectral measurements to the TROPOMI UV ozone profile retrieval.
The effect extends up to an altitude of 30 km. The improvement depends on the latitude and ozone content in the atmosphere. Overall, an improvement of DOF by 1 was observed. In the altitude range of 10 – 20 km, the vertical resolution is about 10 km, which is similar to the values in the stratosphere. The improvement is relatively small in comparison to the results for other combined UV (OMI, GOME-2) and IR (TES, IASI) retrievals obtained in previous publications. We assume that the main reason is the lower spectral resolution of CrIS compared to IASI and TES.

– The validation with tropospheric lidar shows reduced mean differences and reduced standard deviation of the mean differences in tropospheric ozone columns for the UV+IR profile in comparison to the UV-only retrieval. Since only a few tropospheric lidar stations are available, this validation was limited to the northern subtropical region.

– The validation with ozonesondes shows rather minor improvements. When only TOCs are compared, the results from the combined ozone profile retrieval are found to be better than those from UV-only retrieval in the tropics and northern
latitudes. Nonetheless, the UV+IR ozone profiles show a positive bias of +20% to +40% in the altitude range of 10 – 15 km. The reason for this might be due to a positive stratospheric bias in the IR-only retrieval results. In the combined retrieval, the stratospheric bias is removed because of the dominating influence from the UV spectral range. To retain the consistency in the IR spectral region, this is compensated by an over-correction in the 10 – 15 km range, where the sensitivity of the UV measurements is low. A positive stratospheric bias in the IR-only retrieval was also found in
previous publications using TES and IASI data (e.g. Verstraeten et al., 2013; Boynard et al., 2016). Its possible reasons are still a matter of debate.

– Analysing an example one day of collocated MLS and TROPOMI/CrIS measurements, it was shown that the inclusion of the IR spectral range affects the retrieved profiles up to 30 km altitude. In the stratosphere, improvements in comparison with the UV-only retrieval were seen especially in the subtropical region.

There are still some open questions to be answered in the future. The improvement of the combined retrieval over the UV only one was mostly small. One possible reason is a rather low spectral resolution of the CrIS data. It would be therefore interesting to combine TROPOMI data with higher resolved IR instruments, e.g. IASI, however, collocation of TROPOMI with IASI is not as favourable as in the case for the co-flying CrIS instrument. Further investigations are needed to understand the positive bias seen in the stratosphere for the IR-only retrieval. It is expected that the elimination of this bias may help to
further improve the combined retrieval in the troposphere. A potential approach would also be a sequential retrieval where the IR-only retrieval is done first and then used in a 2nd step as a priori for the UV-only retrieval. With respect to the stratospheric ozone profiles further investigations are needed to compare the optimised UV-IR retrieval with optimised UV-only retrieval, i.e. the latter retrieval needs to be performed with its own optimised settings rather than with the same settings as the UV-IR





retrieval. Irrespective of these open questions, it was successfully shown that the approach of a combined TROPOMI and CrIS

ozone profile retrieval is highly promising.



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
