# Peer review of "Combined UV and IR ozone profile retrieval from TROPOMI and CrIS measurements"

_Atmospheric Measurement Techniques, 2021_

## Author Comment (AC1)

The paper is dedicated to retrievals of ozone profiles using combination of UV measurements by TROPOMI and IR measurements by CrIS. The paper describes the retrieval methodology and validation.

The paper is generally well organized and written. Although the improvement from synergy of using TROPOMI and CrIS is rather small, the paper discusses the reasons and a possible way forward. My specific (minor) comments are below.

**We thank the reviewers for their comments and the time spent preparing them.**

COMMENTS

L66 : "However, the IR only retrieval is better than the combined approach in the troposphere". A general comment here (also relevant to the further description and discussion): Is it possible to use only IR retrievals in the troposphere? In other words, one can consider combination of Level 2 profiles by using, for example, a smooth transition to pure IR retrievals in the troposphere. Were there attempts of using such approach?

- **We consider the possibility of combining two L2 profiles as a future approach to be investigated in the outlook of the paper (last paragraph). It could be done, but the difficulty is that there is no straightforward way to combine 2 ozone profiles. The key point here will be the smooth transition you have mentioned.**

L.82: "launch next year" -> It is better to specify the year.

- **done**

L.95: For consistency with Sect.2.1, simply "CrIS" as subsection title would be better.

- **done**

L.103: Please explain the abbreviation CLIMCAPS.

- **done**

L. 118-120: Please explain (very) shortly the meaning of "precision" and "accuracy" (or / and give the reference).

- **The definition and terminology is taken from the MLS user guide. It is clarified in the sentence now.**

L163-164: "The underlying problem is the different SNR of TROPOMI and CrIS and the different spatial resolution of the instrument's measurements from the binning of the pixels". This contradicts with the previous statement in lines 87-88: "The smaller TROPOMI pixels are binned together to match the coarser spatial resolution of CrIS". Please clarify.

- **The corresponding passage was revised:** *"The underlying problem is the different SNR of TROPOMI and CrIS and the different spatial resolution of the instrument's*

*measurements before binning the pixels. The huge and fluctuating differences between the SNR in UV and IR, which are due to the binning and the Illumination conditions, make it nearly impossible to stabilize the retrieval for all possible conditions."*

- **The statement from line 87-88 is correct. Both binned instrument pixels have the same spatial resolution, but due to the binning of the TROPOMI pixels, the UV SNR is much higher than in the IR by pure arithmetic. We assume the problem is that one or both SNR estimations is/are wrong, maybe because of a presence of errors other than a pure random noise, so the fit residuals from both spectral ranges are used for the error covariance matrix.**

L.181-182: Please clarify why both profiles and total column a priori are needed? A general expectation would be that the total column can be obtained from profiles by integration.

- **As shown in the publication, the ozone profile retrieval depends on the a priori ozone profile where the retrieval sensitivity is reduced. Therefore it is important to have an a priori ozone profile as good as possible. We have added the following sentence:** *"The a priori ozone profile originates from a climatology \citep{Lamsal.2004}, where the profile's shape is selected in accordance with the input total ozone value. Additionally, it is scaled with the WFDOAS L2 total column amount \citep{Weber.2018} to receive an a priori ozone profile that is as close as possible to truth."*

L.242: It is better to write "(panel B)"

- **done**

L.265 and below: Day and night tropospheric ozone can be different substantially. However, this is not seen clearly in validation results. Please comment.

- **We have added a comment in the paper:** *"Although daytime and nighttime tropospheric ozone profiles can differ significantly, this is not expected at the Table Mountain station. The station is located at about 1800~m altitude in a non-polluted area and no diurnal variation is expected in the troposphere. Furthermore, the tropospheric lidar does not reach high enough to observe the photolytic diurnal cycle in the upper stratosphere."*

Figure 5. What is the reason for showing the temporal evolution during ~ 1 year? The temporal evolution is not discussed in the paper. Maybe, box-and-whisker plot (without resolving temporal evolution), for each retrieval type, would be more informative?

- **Presenting the comparison as box-and-whisker plot is a great suggestion and we have added those to the figure. However, we would like to keep the time series, as they demonstrate how the measurements of the test dataset are distributed over the entire period and if they show any time dependency. We now refer to these points in the description of the figure.**

[Figure]

*Fig. 1: Absolute differences in the tropospheric ozone content (TOC) with respect to the tropospheric lidar data. The differences are shown as Box-Whisker plots in the left panels and as time series in the middle panels. The right panels show statistical information (mean absolute differences and the standard deviations). TOCs are calculated by integrating ozone profiles from the lowermost retrieval level altitude up to the tropopause. The height of the tropopause is obtained from the ERA5 reanalysis data using the 2PV definition.*

L305: "The results for the other stations are given in the supplement (Fig. S1)". Please say in a few words if the results are the same or different.

- **We have added:** *"They do not show such an impressive improvement of the UV+IR retrieval, as it is seen for Table Mountain, but are in line with the previous assessments from Fig. \ref{fig:lidar_TOC}."*

Figure 8 caption. "The differences to MLS and CrIS data…" Please check the caption and the legends in the figure: there is inconsistency.

- **done**

Figure 9. As in Figure 5, please consider using box-and-whisker plots

- **Here, we have made the same changes as for Fig. 5.**

[Figure]

*Fig 2: Comparison of tropospheric ozone content (TOC) with Box-Whisker plots, the time series and statistical information according to the tropospheric lidar in Fig. 1. Upper panel: Absolute differences in TOC with respect to the ozonesonde data in the tropics (20°S –20°N). Lower panel: same as the upper panel but for northern latitudes (20°N – 60°N).*

Section 5.3 title: Probably, "Comparisons with MLS " would be a better title.

- **done**

L392, misprint in "example"

- **done**

L.412: "…MLS provides the most reliable profiles above 16 km.."  This is true for tropics only. I suggest including the comparison results at least down to the tropopause

- **We have extended the comparison down to 14~km. In the altitudes below, the precision of the individual profiles (which we need) increases up to 100%, as the user guide says.**

Since Sect.6 also contains a discussion, I suggest name this section "Summary and discussion"

- **We think that "conclusion" is a suitable headline. Derived from the concluding remarks  we derive some discussion points and give an outlook.This is typical for a conclusion section.**

Figure 12.  I suggest including also the panel  showing |UV-MLS |  minus |UV& IR – MLS|, where |.| is absolute value  (i.e., difference of  absolute deviations from MLS). Then the regions of improvement will be clearly seen.

- **We adjusted the plots and caption as shown below.**

[revised manuscript text omitted]

fewer collocated pixels are available. From this, the change in the vertical resolution of the profiles could be better understood.

The vertical resolutions of the three retrievals, which are given by the inverted main diagonal of the averaging kernel (AK) matrix, are shown in Fig. 4 (left). This approach is based on the concept of data density (Purser and Huang, 1993) and is explained by the definition of DOF and the resulting assumption that the diagonal of the AK matrix is a "measure of the number of degrees of freedom per level, and its reciprocal is a the number per degree of freedom, and thus a measure of resolution" (Rodgers, 2002, Sec. 3.3, pp. 54). As is known from previous studies, the UV-only retrieval from TROPOMI measurements (blue) has high vertical resolution above 20 km and reduced vertical resolution between 10 and 15 km (Mettig et al., 2021). The IR-only retrieval from CrIS measurements (orange) has a vertical resolution of around 10 km between 5 and 25 km. The combined UV+IR ozone profile retrieval shows a vertical resolution of about 10 km from 5 to 55 km. The contribution from the IR to the combined retrieval diminishes above about 30 km meaning that the upper stratosphere is derived mostly from the UV part of the retrieval. The measurement response functions, shown in the right panel of Fig. 4 confirm the previous findings. In the optimal case the measurement response should approach the unity, which is nearly reached for the combined retrieval

between 10 – 50 km. Below 15 km, the UV-only retrieval shows a lower response than IR and UV-IR retrievals, and above
300   20 km the IR-only retrieval progresses towards zero..

**5   Validation**

The validation of the TOPAS UV+IR retrieval focuses here on the troposphere, which we try to improve using the combined
UV and IR retrieval. Profiles and tropospheric ozone content (TOC) resulting from the TOPAS retrieval are compared with
measurements from ozonesondes and tropospheric lidars. In the stratosphere, the ozone profiles of the combined retrievals
305   largely agree with those from the UV-only retrievals as shown in Sec. 4; the latter has been validated in Mettig et al. (2021).
We only provide some example results in the lower stratosphere.

**5.1   Tropospheric lidar**

For the validation in the troposphere, tropospheric lidar measurements are particularly suitable. There are only three locations
where lidar measurements are carried out regularly with a high temporal frequency (up to two times a day) and with which
310   collocation was found in the TROPOMI test data set period. Since lidars have a high vertical resolution (below 100 m), similar
to the ozonesondes, the lidar altitude grid is adjusted in accordance with Eq. (2) before comparisons are made.

Figure 5 shows the comparison of the TOPAS retrieved ozone profiles and tropospheric lidar measurements at three dif-
ferent sites. While the measurements in Huntsville take place during daytime, the ozone profiles in OHP are measured after
sunset. Only Table Mountain provides night- and daytime measurements where the latter match in time with TROPOMI/CrIS
315   overpasses. Nighttime profiles can reach a height of up to 28 km and are used for comparison up to 25 km into the strato-
sphere. Although daytime and nighttime tropospheric ozone profiles can differ significantly, this is not expected at the Table
Mountain station. The station is located at about 1800 m altitude in a non-polluted area and no diurnal variation is expected in
the troposphere. Furthermore, the tropospheric lidar does not reach high enough to observe the photolytic diurnal cycle in the
upper stratosphere. For each of the stations and each retrieval type, the mean ozone profile in number density, the relative mean
320   difference profile in percent, the standard deviation in percent, and the TOPAS vertical resolution are shown. The AK matrix
can be applied to the re-gridded lidar profiles $x_{\text{coarse}}$ to account for the higher vertical resolution of the lidar measurements.
The vertical convolution with the averaging kernels is done as follows:

$$\hat{x} = x_a + \widetilde{A}(x_{\text{coarse}} - x_a) \tag{4}$$

where $\widetilde{A}$ is the averaging kernel matrix corresponding to the retrieval with transformed variables for relative deviations of the

[revised manuscript text omitted]

---

## Author Comment (AC2)

In this work on the "Combined UV and IR ozone profile retrieval from TROPOMI and CrIS measurements" Mettig et al. provide a first account on the joint retrieval of TROPOMI UV and CrIS IR measurements using their own TOPAS algorithm. This promising work is of scientific value, but its overall presentation and interpretation of results could be improved. Especially regarding the application of averaging kernel smoothing and its effect on the (interpretation of) comparison results the text should be revised.

**We thank the reviewers for their comments and the time spent preparing them.**

Major comments:

Lines 46-48: "While the major challenge for the profiles from UV measurements is the low vertical resolution in the altitude range below 20 km, ozone profiles from IR measurements provide more information about the troposphere, but typically do not retrieve ozone above about 30 km (Bowman et al., 2002)." The physical reasons why this is the case, and hence why a joint retrieval might be beneficial regarding vertical sensitivity, are somewhat missing in the introduction. Please provide a brief discussion and references on the wavelength-dependence of light's atmospheric penetration as a motivation for this work.

- **We have added information about the different sensitivities in the UV and IR ozone retrievals:**

  *"In the UV spectral range, the profile information is derived from the different penetration depths of the short-wave radiation, which works very well at altitudes above the ozone maximum, but worse in the layers near the ground. In the IR range, thermal radiation is emitted by the atmosphere and surface and weakens with the decreasing air density in the upper atmosphere.*

  *The concept of using combined UV and TIR observations to improve the retrieval of vertical profiles of ozone was first discussed…"*

Lines 101-102: "But in comparison to IASI, CrIS has a lower noise. Hence, the ozone information content depending on both, spectral resolution and noise, it should be similar for CrIS and IASI." This is quite a blunt statement that seems to be based on a wild guess only. Please be quantitative, including references, or be more cautious in the formulation, e.g. in terms of "possibly compensating effects".

- **We have removed this statement since we did not find any strict justification for it. In some very limited tests where we applied our retrieval to IASI we found higher information content in IASI data compared to CrIS (approx. 0.5 - 1 more degrees of freedom). This difference is documented in the literature as well (Cuesta et al., 2013; Smith and Barnet, 2020). We therefore believe that this is the main reason for the rather small improvement in the combined retrieval with CrIS.**

Lines 138-143: Better explain that interpolation matrix L has size coarse x fine (therefore requiring a pseudo-inverse) and provides an interpolation form the coarse to the fine grid. More importantly, however, this would also be the place to discuss that you are applying

averaging kernel smoothing as well in your comparisons, with formulas. The current statement on lines 269-270 is too brief and incomplete: Regridding already accounts for a different vertical resolution in terms of representation on a grid; the kernel smoothing induces a vertical convolution, i.e., an effective smoothing over several retrieval levels.

- **We have added the equation which we use to convolve the re-gridded validation profile with the AKs:  x' = x_a + Ã (x_coarse - x_a) and a more detailed description with reference to our previous publication.**

 And most importantly, the statement in lines 271-274 – "where the combined retrieval is sensitive and a single retrieval is not, the former might appear to be worse. This is because the difference between retrieval and the reference profile multiplied by the AK matrix by definition approaches zero in altitude ranges where the retrieval has low sensitivity, i.e. AKs are close to zero." – and its disturbing effect on the further analysis could be avoided: If a distinction is made between using the AK matrix merely as a vertical smoothing matrix on one hand, and the application of averaging kernel smoothing as a method for accounting for retrieval differences on the other hand, this issue does not occur. The first requires a multiplication with a normalized AK matrix (x' = A_normalized x_ref), the second a weighted sum of reference and prior profiles (x' = A x_ref + (I-A) x_prior); see for example Section 4.2 and "averaging kernel smoothing" in Section 4.3.1 of Keppens et al., 2019, respectively.

- **At this point we do not agree entirely. In our understanding, the formula x' = A_normalized x_ref can only be applied without concern if a regularisation with respect to zero a priori profile is performed (x_a = 0). According to the literature (Rodgers 2000, Rodgers and Connor, 2003) we use the general formula for comparison between lower resolved retrieval profiles and finer resolved validation profiles (x' = A x_ref + (I-A) x_prior)**
- **We would like to note that our AK matrix is strictly applicable only to relative difference profiles. We don't see a mathematical argument to apply it directly to profiles.**
- **Of course,  smoothing with the AK matrix can be used as an aproximative approach to account for differences in the vertical resolution, but we do not see any additional benefit to the methods we already use (Rodgers formula, calculation of ozone partial columns).**

Finally, the retrieval 'sensitivity' is often mentioned in the text and used as an explanatory ingredient in the comparison results. From Figure 3 and the accompanying text, however, the total vertical sensitivity is hard to interpret. Please provide integrated vertical sensitivities (AK matrix row sums) in Figure 3, and explain their significance in the text, before using them in the results discussion.

- **We have added a new figure containing the measurement response and the corresponding description to the section:**

  *"The measurement response functions, shown in the right panel of  Fig. … confirm the previous findings. In the optimal case, the measurement response should approach the unity, which is nearly reached for the combined retrieval between 10 -- 50~km. Below 15~km, the UV-only retrieval shows  a lower response than IR and UV-IR retrievals, and above 20~km the IR-only retrieval progresses towards zero."*

[Figure]

*Fig. 1: Left: Vertical resolution of the ozone profiles shown in Fig. \ref{fig:profile_overview} given by the inverse main diagonal elements of the AK matrix. Right: Altitude dependent measurement response functions derived from the sum of the rows of the AK matrix.*

Section 3: The UV and IR retrievals are well explained, but it is less clear how exactly they are combined into a single joint retrieval. An additional few sentences seem to be required on this.

- **Combining both retrievals into a joint retrieval does not contain any further difficulties. To make this clearer, we have included the following passage:**

  *"For the final combination of the two spectral ranges, no additional steps in the Tikhonov regularisation are necessary. In contrast to the individual retrievals, the vector y contains the measurement from both spectral ranges. The forward simulation F(x) is performed according to the following two chapters for both spectral ranges and  the error co-variance matrix $S_y$ is filled with entries for both spectral ranges. All other variables and dimensions remain unchanged."*

Minor comments:

Quantitative results should be included in the abstract, e.g. for the MLS comparisons the mentioning of an "improvement" is insufficient. Results for the ozonesonde validation are even missing in the abstract.

- **We have revised the abstract and added the information about the validation with ozone sondes:** *"The validation of the TOC with ozone sondes has shown that the combined retrieval in the northern latitudes agrees better than the UV-only and IR-only retrieval and also has a lower scatter. In the tropics, the IR-only retrieval provides the best results in terms of TOC. While the TOCs show good agreement in general, the profiles have a positive bias of more than 20\% in comparison to the ozone sondes between 10 and 15~km. The reason is probably a positive stratospheric bias from the IR retrieval."*

Line 24: Provide indicative numbers and/or references for "poorer"

- **Sentence revised into:** *"However, the vertical resolution of the ozone profiles from nadir satellite measurements is coarser by a factor of 3 in the stratosphere (e.g. 2-3km for MLS compared to 6-10km for TROPOMI)."*

Lines 30-34: The goal of this listing is unclear, nor is its intention of being exhaustive or not.

- **Sentence is removed.**

Line 66: Replace "is" by "was found to be" (or something similar) and again add references.

- **done**

Lines 87-88: "The smaller TROPOMI pixels are binned together to match the coarser spatial resolution of CrIS." Provide more details already here, in the form "in x by x bins to x by x km^2"

- **Added the required information:** *"Using the cloud cleared radiance L2 product from CrIS, the spatial resolution ends up being 42$\times$42~km$^2$."*

Line 92: "July 2018 to October 2019" Make clearer that this is also the time range of the data under study.

- **added:** *"and all evaluations in this study are based on data from this period."* **to the sentence.**

Line 104: "in the validation" is unclear. The validation in this work or in the provided reference?

- **added:** *"following validation"*

Line 130: "an excellent option" is not a scientific statement.

- **We exchanged the word excellent and added the reason:** *"valuable option, because of their great vertical resolution and stable and precise ozone profile measurements."*

Lines 150-151: Be more specific on "for instance, the secondary calibration among others"

- **The important corrections within the pre-processing are added:** *"... , for instance, the secondary calibration and the correction for rotational Raman scattering and polarisation."*

Table 1: Some retrieval settings are not discussed in the main text. Briefly mentioning these would be helpful in situating the retrieval.

- **We prefer not to extend the paper by the explanation of the individual settings of the retrieval. A note has been added where the detailed explanations of the settings, that have not changed in comparison to the UV-only retrieval from TROPOMI measurements, can be found.**

  *"The essential retrieval settings for the combined retrieval are listed in Table 1. The settings which remain the same as in the TOPAS UV only retrieval are not explained in detail here. The corresponding information can be found in …."*

Table 1 and lines 169-170: "Above 20 km, the Tikhonov parameter is constant and is 0.02. Below, the values are linearly interpolated between the altitudes 16, 10, 6, and 1 km. Values are: 0.06, 0.1, 0.06 and 0.02, respectively." Please provide some clarification or references on how these values are obtained.

- **As in our previous publication about the TOPAS retrieval (Mettig et al., 2021) the value for the Tikhonov is found by empirical studies. We added the following sentence:** *"The strengths and distribution of the Tikhonov parameter is found through empirical studies by trying to maximise the information content in the retrieval and to minimize the RMS between measurement and forward model. A trade-off between the vertical resolution and stability of the retrieval has to be found."*

Line 182: Provide a reference for the ECMWF ERA-5 reanalysis.

- **done**

Line 221: It is agreed that "This approach represents the most straightforward way to analyse the impact of combining both spectral ranges." but could you, e.g. with reference to Mettig et al., 2021, indicate to what extent the retrievals thus differ from the ideal individual retrieval settings?

- **The main difference between the optimised ozone profile retrieval from Mettig et al. 2021 and the UV-only retrieval in this work is in the vertical resolution in the stratosphere. It is slightly reduced from 6 - 10 km to 7 - 12 km. We have added the following passage:**

  *"The vertical resolution in the stratosphere of the UV-only retrieval, presented here, is somewhat reduced compared to the optimised UV retrieval reported in*

*citet{Mettig.2021}. A compromise has to be made in order to stabilise the lower stratosphere (20 -- 30~km), since both UV and IR measurements affect the ozone profile in this altitude range and any disturbances that may occur have to be compensated for. The following analysis shows that the resolution in the stratosphere is reduced from 6 -- 10~km (optimised retrieval in \citet{Mettig.2021}) to about 7 -- 12~km (UV-only retrieval in this work)."*

Lines 234-235: "The vertical resolutions of the three retrievals, which are given by the inverted main diagonal of the averaging kernel (AK) matrix" Please explain why this is the case and/or provide a reference.

- **We provide the explanation and the reference in the paper now:**

  *"The approach is based on the concept of data density (Purser and Huang, 1993) and is explained by the definition of DOF and the resulting assumption that the diagonal of the AK matrix is a "measure of the number of degrees of freedom per level, and its reciprocal is a the number per degree of freedom, and thus a measure of resolution" (Rodgers, 2002, Sec. 3.3, pp. 54)."*

Lines 251-252: "This may be due to the lower spectral resolution of CrIS compared to IASI and TES." Could this be verified (possibly in future work) by artificially increasing the resolution? Please add a note on this.

- **That is a good approach for future investigations. We have also added the thought of combining TROPOMI and IASI to directly compare both combined retrievals.**

  *"In a future next step, an attempt could be made to artificially increase the spectral resolution of CrIS in a simulation. It is also possible to combine TROPOMI and IASI, even if fewer collocated pixels are available. From the comparison, the change in the vertical resolution of the profiles could be better understood"*

Line 283: "The standard deviations for all comparisons are similar to those of the a priori profile." Please briefly explain what this learns about the data.

- **Added:**

  *" That means, for a single profile, the precision or rather the scattering around the mean value is not improved in comparison to the a priori information."*

Line 290: The term "retrieval layer" suggests rather a retrieval of partial columns, which I understood not to be the case here. Please mention the retrieval units explicitly and possibly change 'layer' to 'level' accordingly.

- **Correctly it should read level. We have corrected that.**

Line 291: Please provide a reference for the "2 PVU definition"

- **We have added a reference for the dynamical tropopause (Hoskins et al. (1985)) and an example where the 2 PUV definition is used for calculating tropospheric ozone columns (e.g. Zbinden et al., 2006)**

Figure 5: It might be more insightful to use the same vertical scale for our four plots.

- **We have generally revised Fig. 5 and ensured that the vertical scales are the same (the same also for the ozone sonde plot).**

Lines 332-334: "The mean collocated ozone profiles from NASA's operational CrIS level 2 product for the same ozonesonde measurements are also shown. For the comparison with CrIS, it must be taken into account that the NASA operational retrieval provides only about 2 DOFs." This dataset is not mentioned in Section 2.3.

- **This dataset is mentioned in chapter 2.2 and the properties and quality characteristics are described in the introduction. We have made sure that Section 2.2 indicates that it is a data set for validation. Since this dataset is of minor importance for the validation, we think that this is sufficient.**

    *"The ozone profile used in the following validation and the surface temperature are taken from the level 2 CLIMCAPS atmosphere cloud and surface geophysical state V2 data product (Barnet, 2019a)."*

Lines 344-345: "Because the UV-only retrieval has a low vertical resolution between 10 and 15 km, it remains close to the climatology." Would you mean low vertical sensitivity here, which seems to be a more appropriate explanation? Please anyhow refer to the latter as well (as is done in the conclusions on lines 448-449), based on the total vertical sensitivity profiles requested in the major comments.

- **We have revised the passage:** *"Because the UV-only retrieval between 10 and 15~km has a low vertical resolution as a result of a low sensitivity , as shown in Fig. …, it remains close to the climatology."*

Editorial:

Line 12: "From the comparison with tropospheric lidars both…" into "In their comparison with tropospheric lidars, both…" → **done**

Line 39: "MetoP" into "Metop" → **removed**

Line 41: Remove "instrument" before the reference. → **removed**

Line 61: "combinations" plural → **done**

Lines 70-76: Adding references to the subsequent sections might help in providing a structured reading.

Line 76: "CrIS" → **done**

Lines 81-82: Be more specific than "next year" → **done**

Line 83: "UVIS" into "UV-VIS" or "VIS" → **done**

Line 173: "Frauenhofer" into "Fraunhofer" → **done**

Caption of Figure 2: "The time difference is 25 minutes…" → **"for" removed**

Figure 3: Add color bar legend. → **done**

Line 373: "received from" into "obtained with" → **done**

Line 388: Correct "is yields" → **"is" removed**

Line 392: "example" → **corrected**

Lines 419-420: "are by about 20% smaller" is a strange formulation → **removed "by"**

Line 452: Rephrase "an example one day" → **done**

[revised manuscript text omitted]

fewer collocated pixels are available. From this, the change in the vertical resolution of the profiles could be better understood.

The vertical resolutions of the three retrievals, which are given by the inverted main diagonal of the averaging kernel (AK) matrix, are shown in Fig. 4 (left). This approach is based on the concept of data density (Purser and Huang, 1993) and is explained by the definition of DOF and the resulting assumption that the diagonal of the AK matrix is a "measure of the number of degrees of freedom per level, and its reciprocal is a the number per degree of freedom, and thus a measure of resolution" (Rodgers, 2002, Sec. 3.3, pp. 54). As is known from previous studies, the UV-only retrieval from TROPOMI measurements (blue) has high vertical resolution above 20 km and reduced vertical resolution between 10 and 15 km (Mettig et al., 2021). The IR-only retrieval from CrIS measurements (orange) has a vertical resolution of around 10 km between 5 and 25 km. The combined UV+IR ozone profile retrieval shows a vertical resolution of about 10 km from 5 to 55 km. The contribution from the IR to the combined retrieval diminishes above about 30 km meaning that the upper stratosphere is derived mostly from the UV part of the retrieval. The measurement response functions, shown in the right panel of Fig. 4 confirm the previous findings. In the optimal case the measurement response should approach the unity, which is nearly reached for the combined retrieval

between 10 – 50 km. Below 15 km, the UV-only retrieval shows a lower response than IR and UV-IR retrievals, and above
20 km the IR-only retrieval progresses towards zero..

**5  Validation**

The validation of the TOPAS UV+IR retrieval focuses here on the troposphere, which we try to improve using the combined
UV and IR retrieval. Profiles and tropospheric ozone content (TOC) resulting from the TOPAS retrieval are compared with
measurements from ozonesondes and tropospheric lidars. In the stratosphere, the ozone profiles of the combined retrievals
largely agree with those from the UV-only retrievals as shown in Sec. 4; the latter has been validated in Mettig et al. (2021).
We only provide some example results in the lower stratosphere.

**5.1  Tropospheric lidar**

For the validation in the troposphere, tropospheric lidar measurements are particularly suitable. There are only three locations
where lidar measurements are carried out regularly with a high temporal frequency (up to two times a day) and with which
collocation was found in the TROPOMI test data set period. Since lidars have a high vertical resolution (below 100 m), similar
to the ozonesondes, the lidar altitude grid is adjusted in accordance with Eq. (2) before comparisons are made.

Figure 5 shows the comparison of the TOPAS retrieved ozone profiles and tropospheric lidar measurements at three dif-
ferent sites. While the measurements in Huntsville take place during daytime, the ozone profiles in OHP are measured after
sunset. Only Table Mountain provides night- and daytime measurements where the latter match in time with TROPOMI/CrIS
overpasses. Nighttime profiles can reach a height of up to 28 km and are used for comparison up to 25 km into the strato-
sphere. Although daytime and nighttime tropospheric ozone profiles can differ significantly, this is not expected at the Table
Mountain station. The station is located at about 1800 m altitude in a non-polluted area and no diurnal variation is expected in
the troposphere. Furthermore, the tropospheric lidar does not reach high enough to observe the photolytic diurnal cycle in the
upper stratosphere. For each of the stations and each retrieval type, the mean ozone profile in number density, the relative mean
difference profile in percent, the standard deviation in percent, and the TOPAS vertical resolution are shown. The AK matrix
can be applied to the re-gridded lidar profiles $x_{\text{coarse}}$ to account for the higher vertical resolution of the lidar measurements.
The vertical convolution with the averaging kernels is done as follows:

$$\hat{x} = x_a + \widetilde{A}(x_{\text{coarse}} - x_a) \tag{4}$$

where $\widetilde{A}$ is the averaging kernel matrix corresponding to the retrieval with transformed variables for relative deviations of the

[revised manuscript text omitted]